# Vasoactive intestinal peptide controls the suprachiasmatic circadian clock network via ERK1/2 and DUSP4 signalling

Ryan Hamnett [1,2], Priya Crosby[1], Johanna E. Chesham[1] & Michael H. Hastings [1]

The suprachiasmatic nucleus (SCN) co-ordinates circadian behaviour and physiology in mammals. Its cell-autonomous circadian oscillations pivot around a well characterised transcriptional/translational feedback loop (TTFL), whilst the SCN circuit as a whole is synchronised to solar time by its retinorecipient cells that express and release vasoactive intestinal peptide (VIP). The cell-autonomous and circuit-level mechanisms whereby VIP synchronises the SCN are poorly understood. We show that SCN slices in organotypic culture demonstrate rapid and sustained circuit-level circadian responses to VIP that are mediated at a cell-autonomous level. This is accompanied by changes across a broad transcriptional network and by significant VIP-directed plasticity in the internal phasing of the cell-autonomous TTFL. Signalling via ERK1/2 and tuning by its negative regulator DUSP4 are critical elements of the VIP-directed circadian re-programming. In summary, we provide detailed mechanistic insight into VIP signal transduction in the SCN at the level of genes, cells and neural circuit.

[1] MRC Laboratory of Molecular Biology, Francis Crick Ave, Cambridge CB2 0QH, UK. [2] Present address: Department of Neurosurgery, Stanford University, 318 Campus Drive, Stanford, CA 94305, USA. Correspondence and requests for materials should be addressed to M.H.H. (email: mha@mrc-lmb.cam.ac.uk)

Circadian (~24 h) rhythms are intrinsic biological oscillations that organise behaviour and physiology into a 24 h programme that adapts an organism to daily environmental cycles. The molecular clockwork driving these rhythms in mammals is a cell-autonomous oscillator, built around a transcriptional–translational feedback loop (TTFL), in which positive factors CLOCK and BMAL1 drive transcription of *Period* and *Cryptochrome*. In turn, PER and CRY feedback to inhibit this transcription by CLOCK:BMAL1. Degradation of PER and CRY dissipates this inhibition and re-initiates the TTFL. Circadian time-keeping across the organism is hierarchical: most cells can maintain cell-autonomous TTFL rhythms but require co-ordination from the hypothalamic suprachiasmatic nucleus (SCN)[1], the central pacemaker. The intrinsic oscillation of the SCN is entrained to the light:dark cycle by direct retinal innervation via the retinohypothalamic tract (RHT), and multisynaptic projections from the SCN in turn synchronise peripheral clocks with the environment and with each other. When isolated in culture, cellular time-keeping in the SCN can persist indefinitely due to powerful circuit-level signalling. An understanding of the mechanistic basis of this signalling is therefore key to understanding the unique pacemaker role of the SCN.

The SCN expresses a diverse array of neuropeptides within regionally specific neuronal sub-populations[2–4]. Cells expressing vasoactive intestinal peptide (VIP) receive retinal innervation and project to strongly rhythmic cells expressing the VIP receptor, VPAC2. VIP signalling via VPAC2 is essential to maintain coherent time-keeping across the SCN network: loss of genes encoding VIP or VPAC2 compromises circadian behaviour and synchrony of the cellular TTFLs of the SCN[5–7]. Moreover, synchronous, high amplitude TTFL oscillations can be restored in VIP-deficient SCN by paracrine signals derived from co-cultured VIP-proficient SCN[8]. VIP/VPAC2 signalling is also important in retinal entrainment of the network. The circadian rhythm of retinorecipient cells (including VIP cells) in the SCN 'core' is proposed to be weakly rhythmic and so readily reset[9–12] via release of glutamate and PACAP from the RHT[13] and subsequent activation of various kinases[14]. These signals converge on $Ca^{2+}$/cAMP response element (CRE)-binding protein (CREB), resulting in CRE-mediated transcription of genes[15,16], including *Per1* and *Per2*[17]. Thus, retinal cues control the TTFL of VIP SCN neurons. These light-dependent changes are then conveyed to, and integrated with, the oscillation of the SCN 'shell'. VIP signalling via the VPAC2 receptor is particularly important for this process[18,19], with responses to light being attenuated in both VIP- and VPAC2-deficient mice[5,20]. Furthermore, application of VIP to SCN slices or injection in vivo acutely phase-shifts rhythms in SCN physiology and behaviour[21–24].

Effective core-to-shell VIP/VPAC2 signalling is therefore critical for two fundamental properties of the SCN: sustained network-level oscillation and photic entrainment. Nevertheless, the underlying mechanisms are poorly understood. VIP may induce *Per1* and *Per2* transcription via pathways involving adenylate cyclase (AC), phospholipase C (PLC) and protein kinase A (PKA)[23,25,26], but deeper understanding of the signalling cascade from VPAC2 activation to circadian gene transcription is lacking. Furthermore, phase shifting of the SCN and behavioural rhythmicity likely involves a complex and multigenic network[27] beyond *Per1* and *Per2*. We therefore undertook a comprehensive characterisation of the molecular mechanisms through which VIP functions. Through application of VIP to SCN slices, we show that its effects on phase, period and amplitude of the circadian TTFL are cell-autonomous, can occur in the absence of an intact SCN network and involve appreciable plasticity within the TTFL. Transcriptomic profiling and pharmacological studies highlight the importance of CRE-mediated transcription in acute VIP signal transduction, particularly via the ERK1/2 cascade. Moreover, DUSP4, a negative regulator of ERK1/2, was identified from microarray analysis as a candidate regulator of the VIP response, and manipulations in vivo and ex vivo demonstrate an important role for DUSP4 in SCN responses to light and VIP. These results thereby reveal signalling pathways that underpin the network-level response of the SCN to VIP.

## Results

**VIP re-programmes the SCN through cell-autonomous mechanisms.** To investigate the effect of VIP/VPAC2 signalling on circadian function, we applied VIP to organotypic SCN slices carrying the PER2::LUC TTFL reporter[28] (Fig. 1a). VIP was delivered at circadian time (CT)10 (two hours before the peak of PER2::LUC, defined as CT12) when VIP has its maximal phase-resetting effect[23]. VIP phase-delayed the SCN circadian oscillation by up to 4 h (Fig. 1a, Supplementary Fig. 1a). This was preceded by acute induction (up to 1.5-fold) of PER2::LUC bioluminescence (Supplementary Fig. 1b), which was highly correlated with the magnitude of the subsequent phase shift (Fig. 1b). VIP also caused a sustained lengthening of circadian period (up to 1.5 h; Supplementary Fig. 1c), and reduced the amplitude of oscillation (up to 90%) (Supplementary Fig. 1d).

Once established, the effects of VIP, particularly on amplitude, could not be reversed by media change: they represent a permanent re-programming of the SCN circuit (Supplementary Fig. 1e). The emergence of these effects was, however, progressive, with reduced amplitude established by only 2 h of VIP treatment, whereas sustained period lengthening required more than 6 h (Supplementary Fig. 1f–j). All effects of VIP were dose-dependent, and comparable dose-dependent effects were observed following treatment with the specific VPAC2 agonist Bay 55-9837[29] at 50 nM or 5 μM (Supplementary Fig. 2a–f).

To compare the phase-shifting action of VIP with that of glutamate, the primary mediator of retinal input to the SCN core, we applied VIP or glutamate via droplet directly on to SCN slices at either CT10 or CT14. Whereas VIP caused significant phase shifts at both times, glutamate caused a phase shift only at CT14 (Supplementary Fig. 2g–i), consistent with previous reports[30,31]. Equally, VIP acutely induced PER2 at both time points but glutamate did so only at CT14 (Supplementary Fig. 2j). Furthermore, glutamate did not result in a subsequently reduced amplitude at either phase, in contrast to VIP (Supplementary Fig. 2k). Thus, the effects of VIP on the SCN clock network are distinct from those of glutamate (and by extension light), in terms of their phase-dependence and molecular consequences[23,31].

Induction of PER2::LUC by VIP was not dependent on a functional TTFL, insofar as bioluminescence in arrhythmic $Cry1^{-/-}Cry2^{-/-}$ (CryDKO) SCN, which lack circadian organisation[8,32,33], exhibited an immediate induction following addition of VIP (Supplementary Fig. 3a–c). Further, VIP application damped and smoothed the bioluminescent trace and significantly decreased the root mean square of PER2::LUC (Supplementary Fig. 3a, b, d), a measure of noise within the bioluminescent signal in CryDKO slices and thus analogous to amplitude in WT slices. Therefore, the molecular cascades whereby VIP acts within VPAC2-positive target cells to control the TTFL can function independently of the TTFL.

VIP could affect SCN rhythmicity by acting at the cellular and/ or circuit levels. A significant feature of circuit-level time-keeping is the spatiotemporal wave of bioluminescence that reflects phase variations in clock gene expression within different regions of the SCN. To characterise the network-level effects of VIP, SCN slices were imaged on CCD camera (Fig. 1c, Supplementary Movie 1) and the spatiotemporal dynamics of PER2::LUC expression were

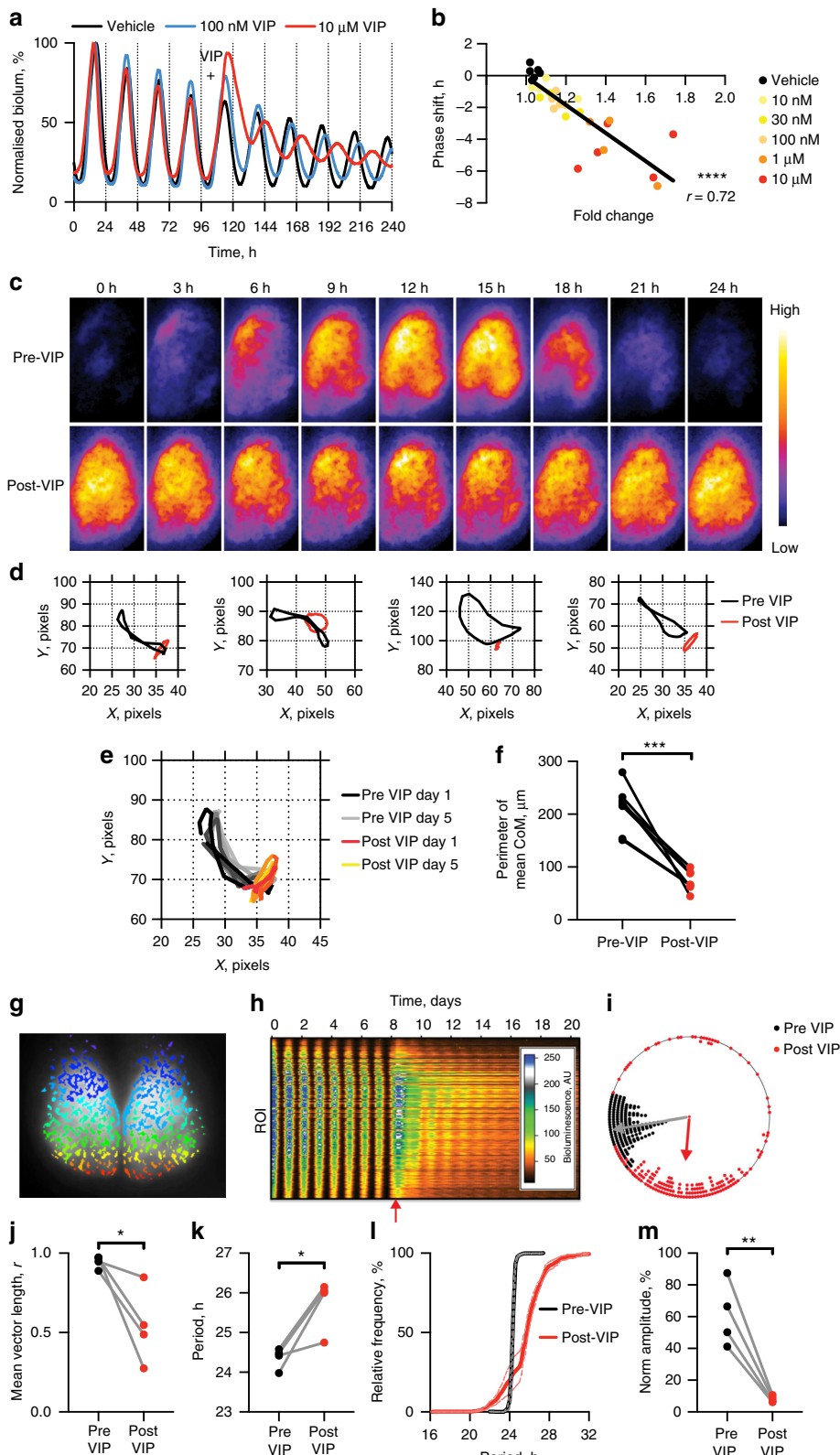

analysed using centre of mass (CoM), which provides an integrated descriptor of the wave[34], and thus the phase relationship between SCN sub-regions. All slices showed a clear and consistent disruption of the spatiotemporal wave immediately after VIP application (Fig. 1d, e), mirroring the effects of VIP cell activation with Gq DREADDS[34]. Not only was the range of the CoM reduced (Fig. 1f), but the directionality of the CoM was

consistently altered from the stereotypical dorsomedial-ventrolateral to a more dorsolateral-ventromedial trajectory after VIP (Fig. 1d, e). This may be in part due to the dorsal tip of the slice displaying a high baseline of bioluminescence but very little oscillation (Fig. 1c). Thus, VIP affects the phase relationships between cellular oscillators within the SCN in a consistent, non-random manner.

**Fig. 1** VIP re-programmes SCN circadian rhythmicity at the cell and circuit level. **a** Representative PER2::LUC bioluminescence rhythms of vehicle-, 100 nM VIP- or 10 μM VIP-treated slices. VIP was bath-applied at CT10 of the 5[th] cycle (marked by plus). Bioluminescence has been normalised to the first peak. **b** Scatterplot of phase shifts vs. PER2::LUC fold induction following VIP treatment. Line represents computed linear regression. $n = 4$–6 slices per concentration of VIP, ****$P < 0.0001$, Pearson's correlation. **c** CCD camera images of PER2::LUC bioluminescence rhythms of representative slice before and after 10 μM VIP. Each set of images has been separately adjusted for maximum contrast. **d** Mean centre of mass (CoM) path vectors for 4 independent SCN slices; means were calculated over 5 days in the given time window. **e** CoM path vectors over time for representative SCN before and after VIP. **f** Group perimeter measurements of CoM path vectors for SCN slices before and with VIP (paired $t$-test, ***$P < 0.001$). **g** Regions of interest (ROI) determined by SARFIA mapped onto phase image of representative SCN. **h, i** Representative raster plot (**h**) and associated Rayleigh plot (**i**) of PER2::LUC bioluminescence rhythms of SCN slices treated with 10 μM VIP imaged on CCD camera (slice in **g**) VIP treatment marked by arrow. **j** Length of Rayleigh plot vector ($n = 4$ slices, mean ± SEM, example in **g–i**) before and after VIP treatment. **k** Periods for all ROIs across the SCN slices ($n = 4$ slices, mean per slice ± SEM; paired $t$-test, *$P < 0.05$). **l** Relative cumulative frequency ($n = 4$ slices, mean ± SEM, %) of ROIs with a given period before and after VIP treatment. **m** Amplitude of all ROIs across the 4 SCN slices (mean per slice ± SEM; paired $t$-test, **$P < 0.01$)

To investigate the contribution of cell-autonomous actions of VIP, individual SCN cells were defined as regions of interest (ROIs, identified using Semi-Automated Routines for Functional Image Analysis (SARFIA)[35] in Igor Pro (Fig. 1g)) and circadian oscillations analysed. VIP had strong effects on the rhythmicity and amplitude of most oscillators (Fig. 1h), abrogating the previously tight phase coherence between cells (Fig. 1i, j). Consistent with the ensemble measures, the majority of ROIs displayed a lengthened period (Fig. 1k, l), and reduced amplitude (Fig. 1m). Thus, exogenous VIP affects cellular TTFLs across the SCN. The reduction in amplitude observed at the network level arises from cell-autonomous effects of VIP as well as network-level phase dispersal, whilst lengthening of ensemble TTFL period is likely cell-autonomous.

To determine whether the effects of VIP at the single cell-level require an intact SCN circuit, slices were treated with tetrodo-toxin (TTX) 24 h prior to VIP. By blocking voltage-gated sodium channels, TTX prevents synaptically mediated intercellular communication, thereby uncoupling the SCN network[36,37]. This was reflected in a reduced PER2::LUC peak prior to VIP addition (Fig. 2a), but VIP nevertheless strongly phase-delayed the ensemble rhythm (Fig. 2a, b), and acutely induced PER2 (Fig. 2c), even though TTX altered the timing of the first PER2 peak following VIP (Fig. 2d). Moreover, TTX did not prevent the sustained period lengthening by VIP (Fig. 2e), nor the VIP-induced reduction in amplitude of the oscillation (Fig. 2f). Interestingly, if VIP was applied to TTX-treated slices at CT22, rather than CT10, no significant phase shifts were observed (Fig. 2g, h), even in the presence of TTX. This mimicked the intact SCN slice response treated at this phase (Fig. 2h–j), including no significant phase shift, suggesting that the known phase-dependence of the phase shift[23] is gated cell-autonomously. In conclusion, the acute and sustained VIP-mediated re-programming is not electrically mediated, and VIP re-programmes the SCN network circadian rhythm through cell-autonomous mechanisms.

**VIP-induced gene networks and cellular signalling pathways**. To identify potential molecular mediators of the effects of VIP on SCN circadian properties, we used microarrays to interrogate the transcriptomes of slices 2 h (CT12) or 6 h (CT16) after treatment with VIP or vehicle (Veh) at CT10. In all, 1165 significantly regulated transcripts were identified, with CT12 VIP vs. CT12 Veh being the comparison with the greatest number (738 genes with $P < 0.05$). The majority (65%) of VIP-regulated transcripts showed elevated expression after 2 h, and they also tended to show a greater fold-change in abundance than did downregulated genes. Of potential interest were clock genes (e.g. *Per1, Per2, Rorα, Dec1*), genes known to be induced in the SCN by light (e.g. *Dusp1, Dusp4*[38,39]), and genes encoding proteins involved in intercellular communication (e.g. *Cartpt, Gjb2*). Downregulated

genes included D-site albumin promoter-binding protein (*Dbp*), a protein regulated by the core circadian TTFL[40].

To aid data visualisation, a heatmap of the 300 genes with the most significant $P$ values (at the CT12 VIP vs. CT12 Veh comparison) was generated (Fig. 3a) using Morpheus software based on globally normalised values. Very little difference could be seen between the CT10 and CT12 Veh groups, whereas samples within the CT12 VIP and CT16 VIP groups showed clear segregation. The majority of selected genes were upregulated 2 h after VIP treatment, with clusters based on either their subsequent decrease (e.g. *Per1* and *Dusp4*), or their maintained increase at CT16.

To validate the results of the microarray, 11 genes were selected for qPCR analysis based on criteria of rhythmic expression and SCN enrichment (Allen Mouse Brain Atlas[41]). The SCN samples were independent of those used for the microarray and an additional CT16 vehicle group was included to aid interpretation. The qPCR data closely reflected the changes observed in the microarrays (Supplementary Fig. 4), and could again be separated into clusters based on response kinetics: the majority of tested genes increased acutely before subsequently declining, with some returning to baseline levels, whereas *Dbp* showed downregulation following VIP, and *Cartpt* and *Vgf* continued to increase at CT16, as did *Per2*, consistent with bioluminescence recordings.

The cellular processes of genes significantly regulated by VIP were interrogated through gene ontology (GO) term analysis with subsequent clustering for visualisation (Fig. 3b). The largest group of transcripts responding after 2 h contained transcriptional regulators. This group was closely connected to genes involved in cellular transduction cascades relating to kinases, particularly MAPK pathways. The relevance of some unexpected clusters, including steroid synthesis, heparin binding and cytokine signalling awaits clarification. Notable significant clusters after 6 h included the down-regulation of previously upregulated transcription factors and MAPK-related transcripts (Fig. 3c).

**VIP transcriptionally regulates genes containing CREs**. To probe for common mechanisms in the transcriptional responses to VIP we focused on CREs, enhancer elements in genes that are bound by transcription factors, such as CREB, and which have a role in the SCN core in light-mediated phase shifting[15,16,42]. Moreover, VPAC2 acts through cAMP- and phospholipase C-signalling[23,26], which could subsequently activate CREs. To identify the extent of CRE-mediated transcription, the VIP-regulated transcripts in the current study were compared to the dataset of Zhang et al.[43]. Chi-squared analysis revealed that for genes upregulated after 2 h and down-regulated after 6 h there was a significant enrichment for CREs within 300 bp of the TATA box (CRE-TATA; Fig. 3d). GO term analysis showed that they were predominantly transcriptional activators and

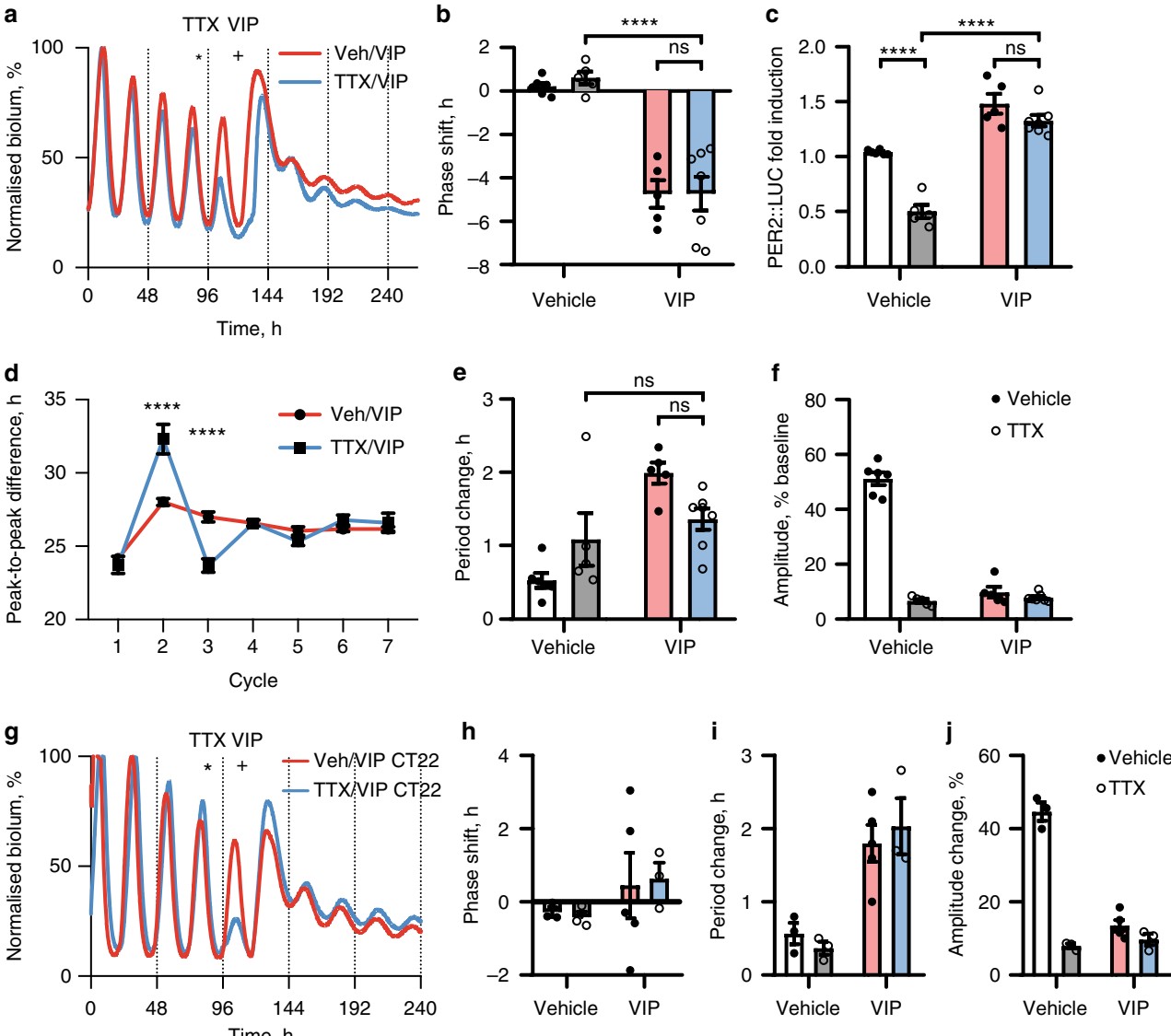

**Fig. 2** VIP re-programmes the electrically silenced SCN network. **a** Representative PER2::LUC bioluminescence rhythms of SCN slices treated with 10 μM VIP at CT10 (marked by plus), after vehicle or 1 μM tetrodotoxin (TTX) 24 h earlier (marked by asterisk). Bioluminescence was normalised to the first peak. **b**, **c** Group data for phase shift (**b**) and acute fold induction of PER2::LUC (**c**) responses relative to predicted values after treatments (mean ± SEM; Veh/Veh (n = 6), TTX/Veh (n = 5), Veh/VIP (n = 5), TTX/VIP (n = 7). **d** Peak-to-peak period following treatments (mean ± SEM). VIP was added between cycles 1 and 2. **e**, **f** Group data for period (**e**) and amplitude (**f**) responses after treatments (mean ± SEM; n as in **b** and **c**. **g** Representative PER2::LUC bioluminescence rhythms of SCN slices treated with 10 μM VIP at CT22 (marked by plus), after vehicle or 1 μM tetrodotoxin (TTX) 24 h earlier (marked by asterisk). **h**–**j** Group data for phase shift (**h**), period change (**i**) and amplitude (**j**) responses after VIP at CT22 preceded by TTX/vehicle (mean ± SEM; Veh/Veh (n = 3), TTX/Veh (n = 3), Veh/VIP (n = 5), TTX/VIP (n = 3). All tests two-way ANOVA with Sidak's (**d**) or Tukey's (**b**, **c**, **e**, **f**) multiple comparisons test, ns = not significant, ****P < 0.0001

MAPK-related transcripts (Supplementary Fig. 5a). This was complemented by the observation that genes downregulated at 2 h and upregulated at 6 h showed no enrichment for CREs (Fig. 3d). Furthermore, genes that were regulated by VIP but did not contain CREs separated out into distinct clusters, for example cytokine-related transcripts lacked CREs (Supplementary Fig. 5a, b). Thus, although not universal, CREs were a significantly enriched marker of responses to VIP.

We tested directly the effect of VIP/VPAC2 signalling on CRE-dependent transcription by transducing SCN slices with a lentiviral vector encoding luciferase under a synthetic promoter containing two CRE sequences (CRE-Luc)[34]. CRE-Luc oscillated prior to treatment (Fig. 4a) and was acutely induced by VIP

(Fig. 4a, b), although in contrast to PER2::LUC there was no measurable change of period (Fig. 4c). We then transduced CRE-Luc slices with an adeno-associated virus (AAV) encoding RCaMP to visualise $[Ca^{2+}]_i$, an important second messenger in CRE regulation, and performed multichannel time-lapse imaging (Fig. 4d; Supplementary Movie 2). Both CRE-Luc and $[Ca^{2+}]_i$ displayed strong oscillations prior to VIP, with the peak of $Ca^{2+}$ characteristically preceding CRE-Luc[34] (Fig. 4e). VIP again triggered a long-lasting induction of CRE-Luc but there was no acute induction of $[Ca^{2+}]_i$, although the baseline increased (Fig. 4d). VIP also altered the circadian phase relationship between CRE-Luc and $[Ca^{2+}]_i$, driving the two signals into antiphase (Fig. 4e, f). Whilst VIP lengthened the period of the

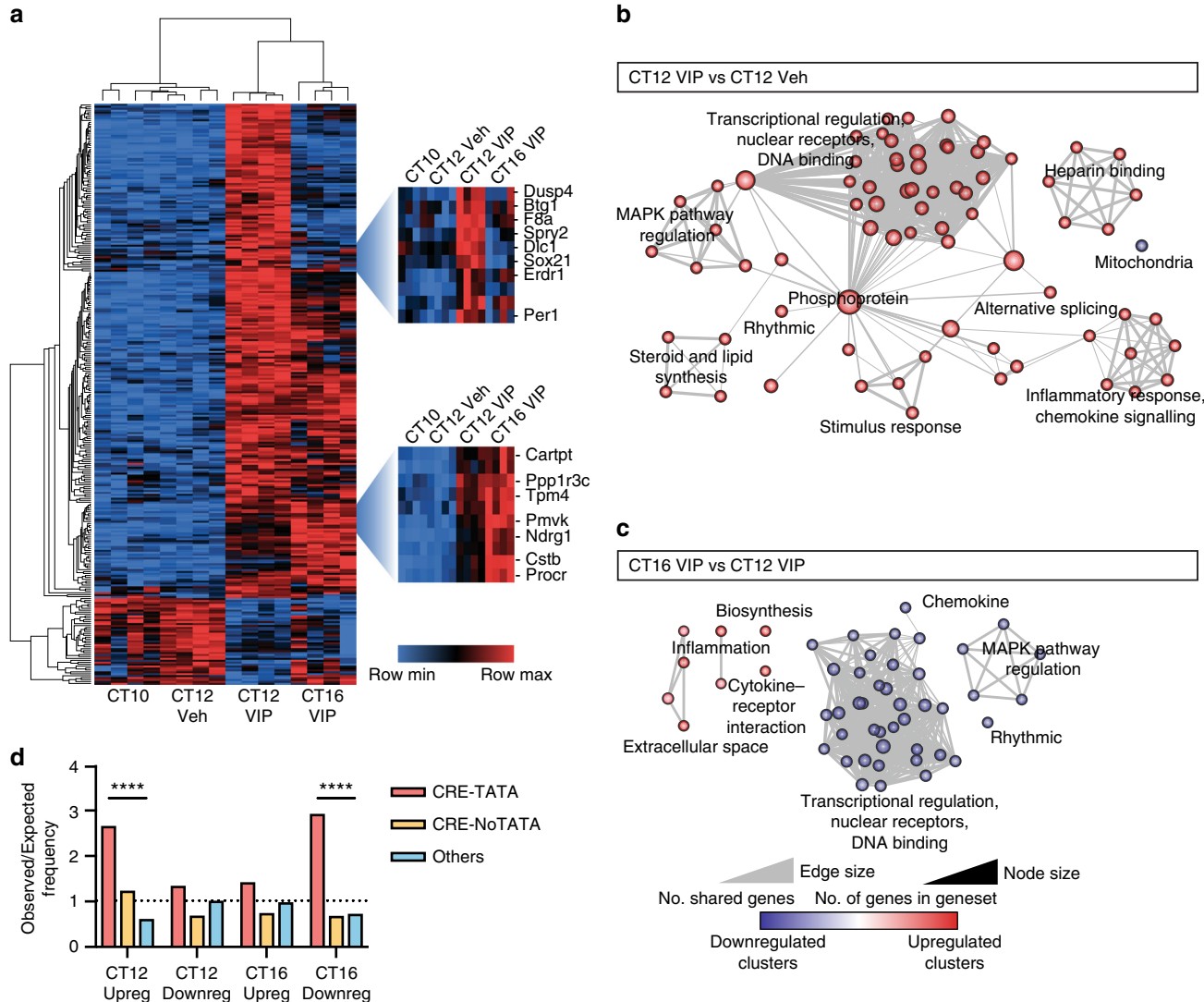

**Fig. 3** VIP-induced gene networks and cellular signalling pathways in SCN. **a** Hierarchically clustered heatmap of top 300 most significantly altered genes in CT12 VIP vs. CT12 Veh comparison across time and treatment using globally normalised probe expression levels. Two regions are enlarged: the top displays genes acutely upregulated 2 h after VIP treatment before returning to baseline levels, the bottom shows genes which continue to be upregulated after 6 h. Within each gene row, transcript levels have been normalised to the minimum (blue) and maximum (red) values for that given gene. **b**, **c** Functional annotation of genes significantly altered by VIP treatment after 2 h (**b**; 738 genes) and after 6 h (**c**; 342 genes) using DAVID functional annotation and visualised using the enrichment map plugin for Cytoscape. **d** Enrichment of CREs in the promoters of significantly regulated genes, displayed as observed/expected frequency, with expected values based on ratios of the appearance of promoter elements across all genes detected on the microarray at CT12 VIP vs. CT12 Veh. CRE-TATA: CRE within 300 bp of TATA box; CRE-NoTATA: CRE in promoter further upstream; Others: No CRE identifiable in promoter (promoter defined as 3 kb upstream to 300 bp downstream of the transcription start site). Chi-squared test, ****$P < 0.0001$

$[Ca^{2+}]_i$ oscillation (as with PER2::LUC), there was no measurable change in period of the CRE-Luc rhythm (Fig. 4g), even though VIP reduced the amplitude of both CRE-Luc and $[Ca^{2+}]_i$ oscillations (Fig. 4h, I), as with PER2::LUC.

**TTFL components are re-programmed by VIP**. We tested whether the re-aligned rhythms of CRE-transcription and $[Ca^{2+}]_i$ led to differential effects on core genes of the TTFL, given the known differences in the number and responsiveness of CREs within their respective promoters[17,43]. Vehicle or VIP was applied at CT10 to SCN slices carrying either a Cry1-Luc[44], Per1-Luc[45] or PER2::LUC reporter (Fig. 4j). Their responses were markedly different. PER2::LUC displayed acute induction, delayed phase, increased baseline and reduced amplitude. In contrast, the induction of Per1-Luc presented as an increased baseline, but without an associated suppression of peak bioluminescence. These different induction profiles for *Per1* and *Per2* were confirmed by qPCR (Supplementary Fig. 4). Cry1-Luc did not show an acute induction, consistent with the absence of CREs in the reporter, although the peak was broadened following VIP. The amplitude was reduced, but the baseline did not increase. Interestingly, PER2::LUC and Cry1-Luc stayed in phase following their VIP-induced phase shifts, whereas Per1-Luc phase shifted significantly less (Fig. 4j, k). Finally, the period of all reporters lengthened comparably (Fig. 4l). Thus, VIP signalling re-programmed the normally tight phase relationship of components of genetic (CRE, TTFL) and cytosolic ($Ca^{2+}$) oscillations, revealing significant VIP-directed plasticity in the cell-autonomous TTFL.

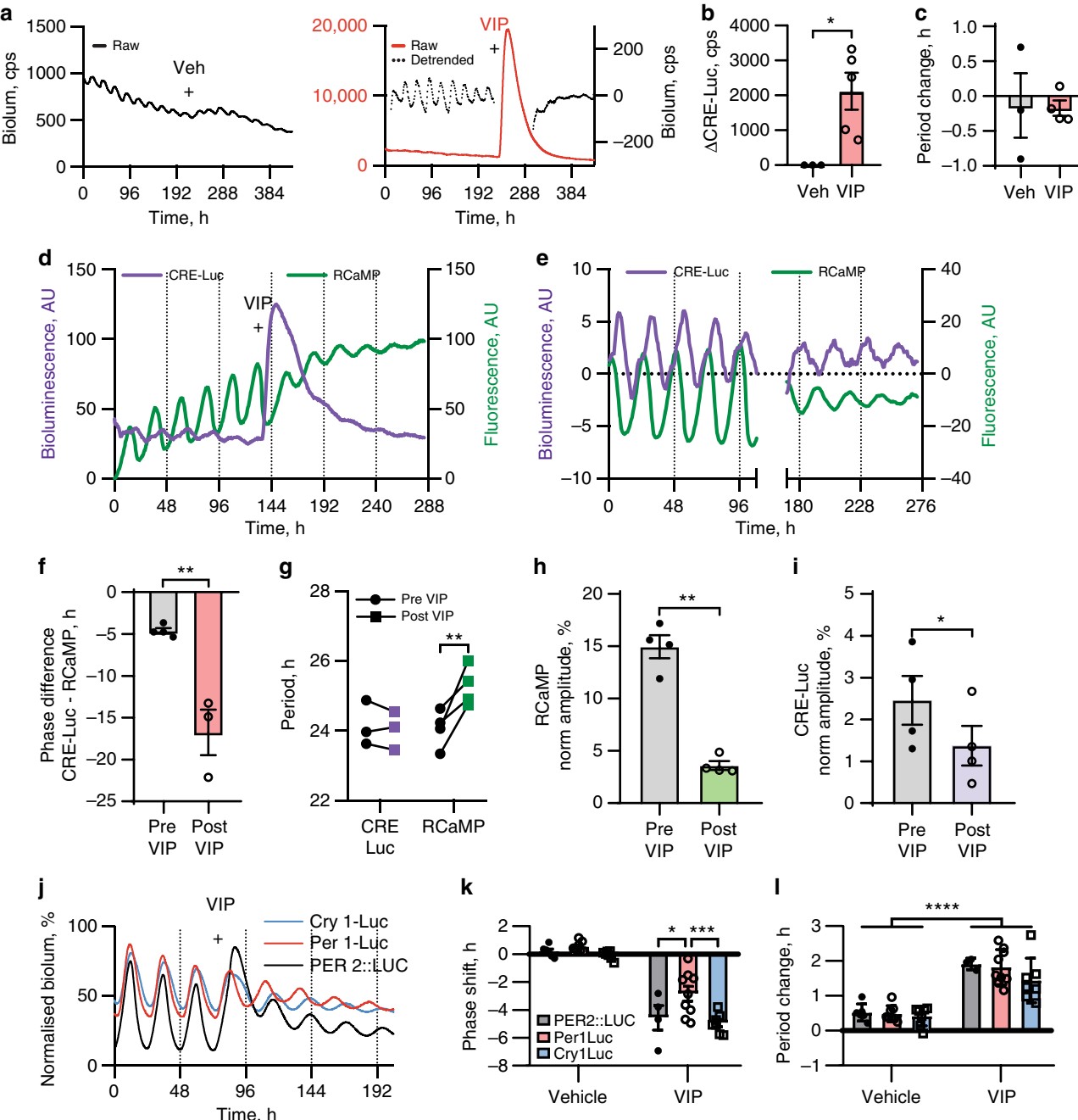

**Fig. 4** VIP results in CRE activation and re-programmes the TTFL of the SCN. **a** Representative CRE-Luc bioluminescence rhythms of SCN slices treated with vehicle (left) or 1 μM VIP (right; treatments marked by plus). A detrended trace is included with the VIP-treated slice to aid visualisation of the oscillation. **b**, **c** Group data for acute induction of CRE-Luc bioluminescence (**b**) and period change (**c**) following treatment with vehicle or VIP (mean ± SEM, veh, $n = 3$; VIP, $n = 4$, unpaired $t$-test, *$P < 0.05$). **d** Representative CRE-Luc bioluminescence and RCaMP Ca$^{2+}$ rhythms from SCN slice treated with VIP ( + ). **e** Detrended CRE-Luc bioluminescence and RCaMP Ca$^{2+}$ rhythms before and after VIP (slice as in **d**). Note altered phase relationship following VIP. Rhythms were detrended by 24 h rolling average subtraction, and have been offset on the y-axis to aid visualisation. **f**, **g** Group data for phase of CRE-Luc rhythm relative to RCaMP rhythm (**f** negative values where CRE-Luc peaks later than RCaMP, mean ± SEM, $n = 3$) and period comparison between CRE-Luc and RCaMP, before and after VIP treatment (**g** two-way ANOVA with Sidak's multiple comparisons test, **$P < 0.01$). **h**, **i** Group data for amplitude changes of CRE-Luc (**h**) and RCaMP (**i**) rhythms before and after VIP treatment (mean ± SEM, $n = 3$, paired $t$-test, *$P < 0.05$, **$P < 0.01$). Data were normalised to the highest value in the recording. **j** Representative PER2::LUC, Per1-Luc and Cry1-Luc bioluminescence rhythms of SCN slices treated with 1 μM VIP (at CT10, marked by plus). Bioluminescence was normalised to the first peak of the recording (not shown). **k**, **l** Group data for phase shift (**k**) and period change (**l**) responses (mean ± SEM) to VIP applied to either PER2::LUC ($n = 4$), Per1-Luc ($n = 9$) or Cry1-Luc ($n = 7$) slices, with vehicle controls (PER2::LUC: $n = 6$; Per1-Luc: $n = 8$; Cry1-Luc: $n = 7$; two-way ANOVA with Sidak's multiple comparisons test, *$P < 0.05$, ***$P < 0.001$)

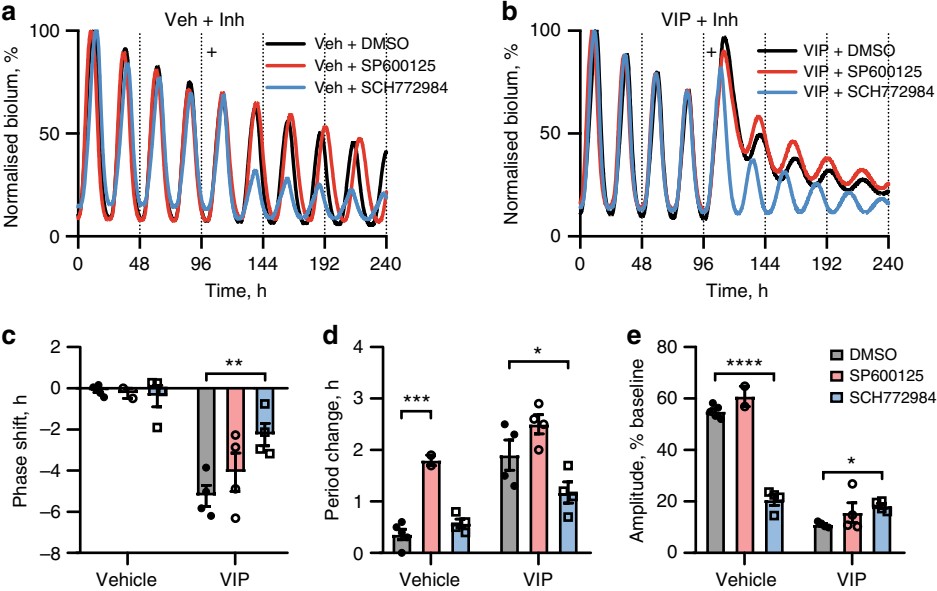

**Fig. 5** ERK1/2 activation is essential for the VIP response. **a**, **b** Representative PER2::LUC bioluminescence rhythms of SCN slices treated with vehicle (**a**) or 1 μM VIP ((**b**) marked by plus) at CT10, 30 min after 3 μM SP600125 (JNK1/2/3 inhibitor), 100 nM SCH772984 (ERK1/2 inhibitor) or vehicle. Bioluminescence has been normalised to the first peak. **c–e** Group data for phase shift (**c**), period change (**d**) and amplitude (**e**) responses after treatments (mean ± SEM; Veh/Veh (n = 5), Veh/VIP (n = 4), SP600/Veh (n = 3), SP600/VIP (n = 4), SCH772/Veh (n = 4), SCH772/VIP (n = 4); two-way ANOVA with Dunnett's multiple comparisons test, *P < 0.05, **P < 0.01, ***P < 0.001)

**ERK1/2 activation is essential for the VIP response**. To explore cellular mediators of this plasticity, we focussed on the GO term cluster analysis, which strongly implicated the MAPK pathway, regulators of which were acutely induced by VIP. To test this directly, PER2::LUC SCN slices were treated with inhibitors to the principal arms of the MAPK pathway, JNK1/2/3 (3 μM SP600125) and ERK1/2 (100 nM SCH772984), followed by vehicle or VIP (Fig. 5a, b). Neither drug resulted in a significant phase shift alone (Fig. 5a, c), but inhibition of JNK1/2/3 increased period (Fig. 5a, d), while ERK1/2 inhibition reduced the amplitude of oscillation (Fig. 5a, e), highlighting independent roles for these two arms of the MAPK pathway in the free-running TTFL. JNK1/2/3 inhibition had no impact on the responses to VIP, whereas ERK1/2 inhibition greatly attenuated them (Fig. 5b), reducing the VIP-induced phase shift, period lengthening and amplitude reduction (Fig. 5c–e). Neither inhibitor affected acute induction of PER2::LUC (Supplementary Fig. 6a). Further evidence for ERK1/2 involvement came from immunostaining SCN slices, whereby VIP treatment at CT10 increased phosphorylated ERK1/2 (pERK) signal (Supplementary Fig. 6b, c). Moreover, by exploiting intersectional expression of TdTomato fluorescent reporter in a *Vpac2-Cre* mouse line, VIP-induced pERK signal could be clearly localised to VPAC2-positive cells (Supplementary Fig. 6d).

Brain-derived neurotrophic factor (BDNF) is a known activator of the ERK1/2 pathway and a component of circadian phase-resetting in response to light[46]. Treatment of PER2::LUC slices with BDNF, however, elicited a response that contrasted with that of VIP (Supplementary Fig. 7a), displaying a phase advance, no period change, and an amplitude increase (Supplementary Fig. 7b-d), with the only commonality between the responses being an increased baseline of PER2::LUC. To confirm further the specific contribution of the ERK1/2 pathway to the VIP response, the involvement of two other kinases frequently implicated in circadian phase-resetting, protein kinase A (PKA)[25] and protein kinase C (PKC)[47], was tested pharmacologically (PKA: 1 μM PKI 14-22 amide myristoylated or 50 μM Rp-8-Br-

cAMPS; PKC: 300 nM sotrastaurin). These inhibitors had no significant effect on the response to VIP when applied individually (Supplementary Fig. 7e-h), but when PKI 14-22 and sotrastaurin were applied in combination, they potentiated the phase-shifting (Supplementary Fig. 7e) and fold-induction (Supplementary Fig. 7f), a result not reproduced using Rp-8-Br-cAMPS in place of PKI 14-22. Finally, the observed enrichment of CREs in the promoters of VIP-regulated transcripts suggested that CREB could be involved in the VIP response, potentially downstream of ERK1/2 activation. We therefore applied VIP in the presence of CREB inhibitor 666-15[48]. 666-15 increased period when applied alone (Supplementary Fig. 7g), indicative of a role for CREB in the ongoing TTFL, but, surprisingly, had no impact on any parameter of the response to VIP (Supplementary Fig. 7e-h). In conclusion, ERK1/2 is critical for VIP signalling, but by specific pathways independent of PKA, PKC and CREB.

**Dusp4 regulates circadian responses to light in vivo**. Confirmation of ERK1/2 involvement directed attention to VIP-responsive transcripts related to the MAPK pathway. These included negative regulators belonging to the dual specificity phosphatase (DUSP) family, including *Dusp1*, *Dusp4*, *Dusp10* and *Dusp14*. Due to its strong upregulation after VIP, pronounced SCN localisation (Allen Mouse Brain Atlas[41]), rhythmic mRNA expression (CircaDB[49]), close clustering with *Per1* responses to VIP (Fig. 4a) and induction by light in the SCN core and shell[27,39], we focussed on *Dusp4*, which can dephosphorylate MAP kinase proteins such as ERK1/2 and JNK1/2/3[50], as a potential regulator of the effects of VIP.

We first assessed the circadian relevance of DUSP4 by recording wheel-running behaviour in mice carrying a DUSP4 'knockout-first' allele, which contains a targeted insertion of a LacZ reporter into the *Dusp4* gene[51]. X-gal staining revealed expression of the allele in the SCN, dentate gyrus, cerebral cortex and piriform cortex (Supplementary Fig. 8a). Mice with *Dusp4*[+/+], *Dusp4*[+/−], and *Dusp4*[−/−] genotypes had normal, robust wheel-running rhythms

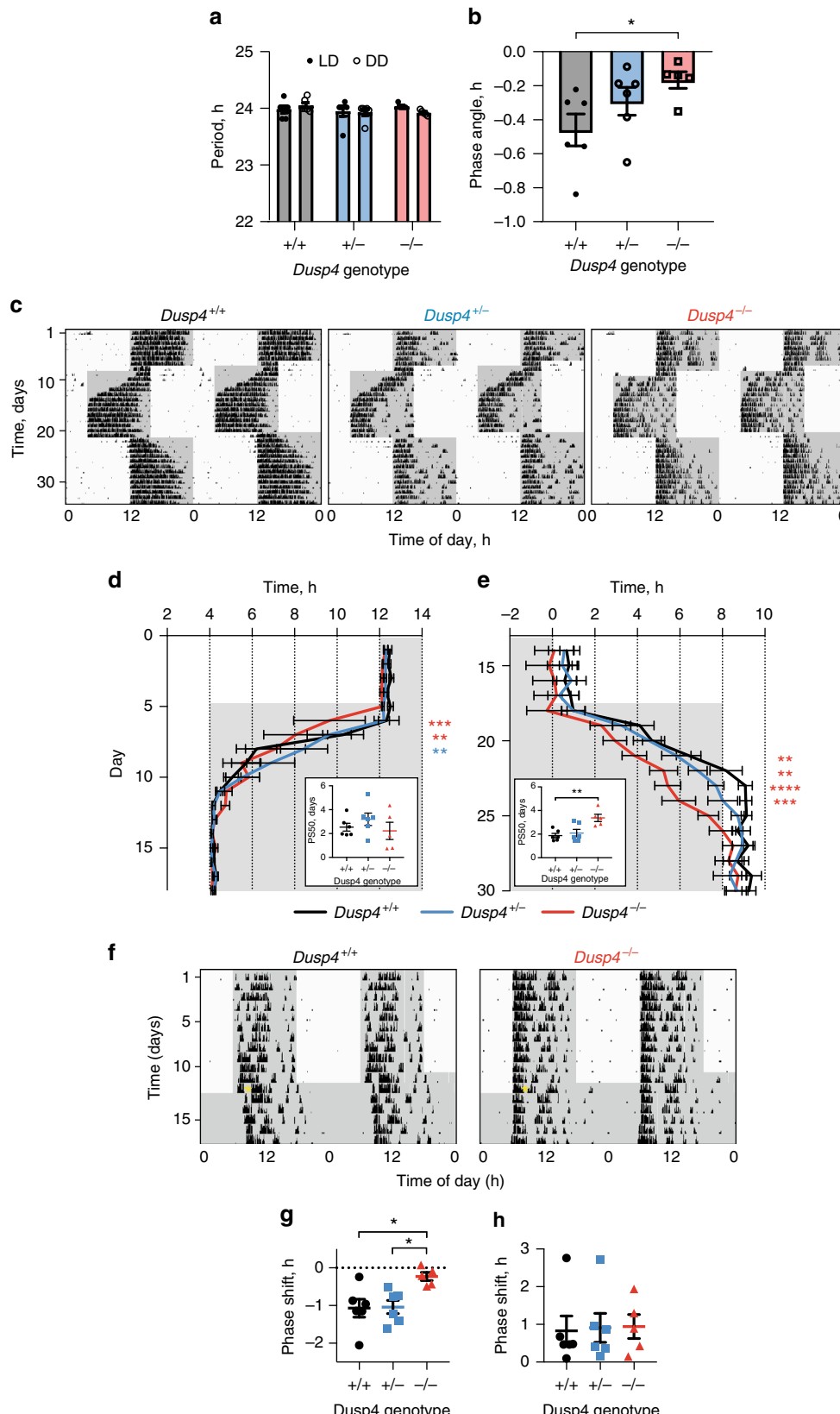

in 12:12 LD and DD (Supplementary Fig. 8b), and no significant differences in period (Fig. 6a) or activity duration (alpha; Supplementary Fig. 8c), although $Dusp4^{-/-}$ mice had a significantly shorter phase angle of entrainment (~18 min) (Fig. 6b), consistent with an attenuated response to light.

The mice were then exposed to experimental 'jet-lag' conditions (Fig. 6c). The 12:12 LD lighting was phase-advanced by 8 h, resulting in a gradual resetting of behavioural onset to realign with 'lights off'. $Dusp4^{-/-}$ behavioural onsets advanced faster in the first two days following the switch (Fig. 6d), although there

**Fig. 6** Mice lacking Dusp4 exhibit attenuated circadian responses to light in vivo. **a**, **b** Group data for circadian period of wheel-running activity rhythms in 12:12 light:dark (LD) and continuous dim red light (DD) (**a**) and phase angle of entrainment (**b**) of $Dusp^{+/+}$ ($n = 6$), $Dusp4^{+/-}$ ($n = 6$) and $Dusp4^{-/-}$ ($n = 5$) mice (mean ± SEM). Phase angle refers to time from lights off until activity onset. Negative values indicate activity onset after lights off. **c** Representative double-plotted actograms of wheel-running behaviour of $Dusp^{+/+}$, $Dusp4^{+/-}$ and $Dusp4^{-/-}$ mice exposed to an 8 h phase advance and an 8 h phase delay in LD cycle. Grey shading represents dim red light. **d**, **e** Group data for activity onsets during the phase advance (**d**) and activity offsets during the phase delay (**e**) of $Dusp^{+/+}$, $Dusp4^{+/-}$ and $Dusp4^{-/-}$ mice (mean ± SEM). Two-way ANOVA with Dunnett's multiple comparisons test. Inset: 50% phase shift values (PS50; mean ± SEM) of phase advances and phase delays, one-way ANOVA with Dunnett's multiple comparisons test. **f** Representative double-plotted actograms of wheel-running behaviour of $Dusp^{+/+}$ and $Dusp4^{-/-}$ mice exposed to a 30-minute phase delaying light pulse (yellow asterisk) at ZT14 after 12:12 LD, then immediately placed into DD. Grey shading represents dim red light. **g**, **h** Group data for phase delays (**g**) and phase advances (**h**) of wheel-running activity of $Dusp^{+/+}$, $Dusp4^{+/-}$ and $Dusp4^{-/-}$ mice following light pulses at CT14 and CT22, respectively. One-way ANOVA with Tukey's multiple comparisons. *$P < 0.05$, **$P < 0.01$, ***$P < 0.001$, ****$P < 0.0001$

was no significant difference in the total time taken to readjust, as evidenced by the 50% phase shift (PS50) value (Fig. 6d inset). In contrast, $Dusp4^{-/-}$ offsets delayed significantly more slowly to an 8 h delay (Fig. 6e), taking 2 days more to reach PS50 (Fig. 6e inset). Consistent with these directionally specific effects in jet-lag, phase delays in response to a single light pulse at ZT14 were almost completely abolished in $Dusp4^{-/-}$ mice (Fig. 6f, g). In contrast, no significant difference between genotypes was observed for an advancing light pulse (30 min, 400 lux) delivered at ZT22 (Fig. 6h), although advances for all genotypes at this phase were not robust. To obtain a molecular correlate of this behavioural effect, pERK levels were determined in the SCN of control and $Dusp4^{-/-}$ mice immediately following a delaying 30-minute light pulse at ZT14. This revealed an attenuated induction of pERK signal (by ~20%) in the mutant mice (Supplementary Fig. 8d, e), consistent with their reduced behavioural sensitivity to light.

**Dusp4 CRISPR knockout potentiates the VIP response.** These in vivo experiments suggested that DUSP4 is an important regulator of input to the circadian system, but it is difficult to separate any role within the retinorecipient cells responding to RHT-derived glutamate and PACAP, from a role in more distal responses to VIP released by light-activated VIP neurons. We therefore investigated DUSP4 in the context of VIP application to SCN slices. DUSP4 was deleted using two AAVs encoding the CRISPR-Cas9 system[52]: one encoded the Cas9 protein under the Mecp2 promoter (Mecp2.SpCas9), while the second (U6.Dusp4g1.hSyn.GFP-KASH) encoded a guide RNA (gRNA) sequence (Dusp4 g1). This approach by-passes potential developmental defects or compensatory mechanisms resulting from traditional genomic knockout of DUSP4. Expression of these vectors resulted in undetectable levels of DUSP4 in N2A cells and SCN slices (by western blot and qPCR, respectively; Supplementary Fig. 9a, b). SCN transduced with both AAVs ($Dusp4^{CRISPR}$) had a significantly shorter circadian period compared to AAV Cas9-only slices (Fig. 7a, c), indicative of a role for Dusp4 in modulating the rate of the steady-state oscillation. Furthermore, deletion of Dusp4 amplified the response to VIP: $Dusp4^{CRISPR}$ slices were phase-delayed by ~30 min more (Fig. 7b, d) and the period increase was greater by ~0.7 h (Fig. 7e). To complement this result, we used an alternative method of depleting DUSP4 levels (Fig. 7f), whereby slices from $Dusp4^{flx/flx}$ mice were treated with an AAV encoding Cre recombinase under the neuron-specific synapsin promoter (hSyn-Cre). This again potentiated the VIP-induced period increase, by ~1.1 h more. Thus, two independent approaches that compromised Dusp4 expression resulted in an amplified VIP response, consistent with its role as a negative regulator of VIP-mediated ERK signalling.

**Constitutive DUSP4 expression reduces the VIP response.** As a complementary approach to knocking out/down, DUSP4 was

constitutively overexpressed in neurons of SCN slices using two AAVs: one encoded Dusp4 cDNA in double-floxed inverted orientation (DiO) under the Ef1α promoter (Supplementary Fig. 10a; $DUSP4^{Ox}$) and the second was hSyn-Cre AAV for neuronal targeting (generating $SynCre-DUSP4^{Ox}$ slices). The $DUSP4^{Ox}$ vector was validated in N2A cells, where DUSP4 expression was shown to be Cre-dependent and predominantly localised to the nucleus (Supplementary Fig. 10b, c). $SynCre-DUSP4^{Ox}$ slices (Fig. 8a) had a shorter steady-state period (~0.5 h) compared to SynCre or $DUSP4^{Ox}$ slices (Fig. 8b, Supplementary Fig. 10d). Following VIP treatment (Fig. 8c), there was no significant difference in acute PER2 induction or phase shifting (Supplementary Fig. 10e, f), but the period change and amplitude reduction were both reduced in $SynCre-DUSP4^{Ox}$ slices compared to controls (Fig. 8d, e). To focus on a role for DUSP4 in VPAC2-positive cells alone, DUSP4 was overexpressed in Vpac2-Cre SCN slices (Fig. 8f, g). Unlike pan-neuronal expression, expression of DUSP4 in VPAC2 neurons did not affect steady-state period (Supplementary Fig. 10g,h), and it had no significant effect on acute PER2 induction or phase shifting following VIP application (Supplementary Fig. 10i,j). Amplitude reduction was reduced relative to $DUSP4^{Ox}$ slices, albeit not compared to VpacCre controls (Fig. 8h). Overexpression of DUSP4 specifically in VPAC2-positive cells did, however, significantly attenuate the VIP-induced period lengthening (Fig. 8i). Taken together, these deletion and overexpression experiments highlight DUSP4 as a negative regulator of the SCN slice response to VIP, with a role in neurons across the SCN, including VPAC2-positive cells in terms of resetting ensemble circadian period.

## Discussion

The retinorecipient VIP-expressing cells of the SCN perform two vital circadian functions: they maintain synchronised circadian oscillations across the SCN in the absence of entraining cues and, by photically regulated release of VIP, they integrate light-dark cycle cues with the intrinsic oscillation of the SCN network. Through a combination of pharmacology and microarray analysis, we assessed the molecular mechanisms through which VIP acts, and revealed a pronounced VIP-directed plasticity in SCN circadian time-keeping at the levels of both the cell-autonomous TTFL and the network. We have identified the ERK1/2 pathway as an essential mediator of the effects of VIP on the SCN and highlighted the role of DUSP4, a negative regulator of MAPK pathways, in this re-programming, with DUSP4 tuning the SCN response to both light in vivo and VIP ex vivo.

VIP, a VPAC2 agonist, and glutamate, the retinal transmitter, were able to shift the ensemble phase of the SCN TTFL. In all cases, the size of the phase delay was positively correlated with the magnitude of acutely induced PER2, highlighting Per gene induction as an entry point to the TTFL. Importantly, however, the actions of VIP and VPAC2 agonist were distinct from those of glutamate in terms of their phase-dependence and the

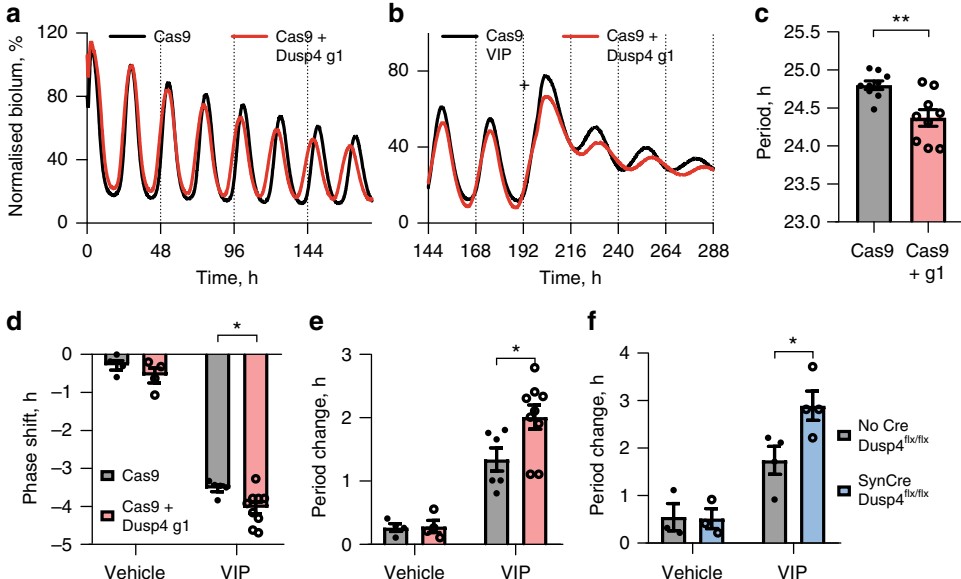

**Fig. 7** *Dusp4* knockout shortens circadian period and potentiates the VIP response of SCN slices. **a**, **b** Representative PER2::LUC bioluminescence rhythms of slices transduced with *Mecp2.SpCas9* alone or in addition to *U6.Dusp4g1.hSyn.GFP-KASH* before treatment (**a**) or with 1 µM VIP application (**b** marked by plus). Bioluminescence has been normalised to the first peak. **c** Group data for steady-state period of slices transduced as in (**a**) (mean ± SEM, $n = 9$ per group, unpaired $t$-test). **d**, **e** Group data for phase shifts (**d**) and period changes (**e**) following vehicle or VIP of slices transduced as in **a** (mean ± SEM, $n = 4$ for all vehicle-treated groups, $n = 6$ for Cas9 with VIP, $n = 9$ for Cas9 + Dusp4 g1 with VIP). **f** Group data for period change of *Dusp4^flx/flx* SCN slices transduced with *hSyn.GFP::Cre* AAV and treated with vehicle or VIP, along with non-transduced controls (mean ± SEM, $n = 3$ for all vehicle-treated groups, $n = 4$ for all VIP-treated groups). Two-way ANOVA with Sidak's multiple comparisons. *$P < 0.05$, **$P < 0.01$

re-programmed molecular response of the TTFL. VIP activity does not, therefore, mimic the effects of light pulses and glutamate, although the period and amplitude effects of pharmacological treatment with VIP are reminiscent of constant light (LL) effects in vivo[53]. It is recognised that the bath application of VIP to slices used here is a simplified system that may not reflect all aspects of endogenous VIP signalling, such as duration of VIP exposure. Nevertheless, previous studies have revealed very clearly an important role for diffuse, paracrine signalling of VIP[8], and when VIP was applied here for shorter time periods, its effects (PER2 induction, phase shifting, amplitude reduction) were still observed and developed progressively. Our TTX experiments suggested that the effects of VIP are likely mediated at the cell-autonomous level, leading us to examine intracellular signalling cascades to map how VIP regulates circadian functions.

Analysis of transcriptional responses to VIP revealed informative similarities and differences between VIP-regulated genes in the current dataset and transcriptome profiles obtained following light pulses in vivo[27,39,54,55]. Notably, immediate early genes, including *Fos* and *Egr1*, as well as *Per1/2* and *Dusp* genes, are acutely upregulated in both conditions, and may represent common cellular responses to input stimuli. Many of these genes also contain CREs within their promoters, further demonstrating commonality between VIP- and light-induced genes. Our pharmacological evidence, however, suggests that CREB itself is not involved in the response to VIP: indeed, photic induction of pCREB occurs only in the core, not the shell, SCN[16,56]. Likely, other transcription factors that act at CREs, such as cAMP responsive element modulator (CREM), or an activating transcription factor (ATF) protein such as ATF3, both of which were upregulated in our microarray, may be important. A further interesting difference between light- and VIP-regulated genes is that light downregulates many genes[39], while VIP was predominantly upregulatory. Moreover, a number of key light-induced regulators, such as *Sik1*[39], did not respond to VIP. The SCN therefore exhibit overlapping but distinct transcriptional

programmes in response to either light or VIP. Furthermore, in contrast to previously reported regulation of AVP by VIP[22,57], no canonical SCN neuropeptides (e.g. AVP, GRP, Prok2) were found to be acutely regulated by VIP. There was, however, a strong upregulation of the neuropeptides *Vgf*[58] and *Cartpt* after 2 h, and *Galanin* after 6 h.

VIP can directly influence TTFL transcriptional regulation, as confirmed here by VIP-mediated upregulation of *Per1* and *Per2*[57,59,60]. Our observation that VIP also regulates genes of auxiliary TTFL loops, such as *Rorα*[61] and *Dec1*[62,63], suggests that its effect on the cell-autonomous oscillator may be more extensive. Whether these responses all occur together in individual cells, or whether VIP induces cell-specific programmes of circadian gene expression awaits determination.

Notwithstanding their common activation, *Per1* and PER2 responded to VIP with interesting differences. Both showed a phase shift following VIP addition, but the exact waveform kinetics and the extent of shifting differed between them. Although we cannot exclude an effect of different reporter kinetics (specifically translation vs. transcription), the fact that qPCR profiles after VIP for *Per1* and *Per2* were also distinct strongly supports the notion that VIP differentially regulates these two TTFL components, leading to an altered phase relationship. Mechanistically, this may be a consequence of altered CRE regulation: *Per1* is more strongly regulated by CREs than *Per2*[17], and a separation of both phase and period between CRE-Luc and Ca$^{2+}$ rhythmicity was observed within individual slices after VIP. This suggests that at least two different phase relationships between TTFL components are capable of sustaining stable molecular oscillations. It is likely that SCN encoding of photoperiod for seasonal responses would be one realistic setting in which multiple context-specific configurations may be exploited by the SCN. Indeed, it has been suggested that the desynchronisation between cells following VIP addition ('phase tumbling'), as seen in the current study, may serve to enhance the ability of external cues to entrain rhythms at the

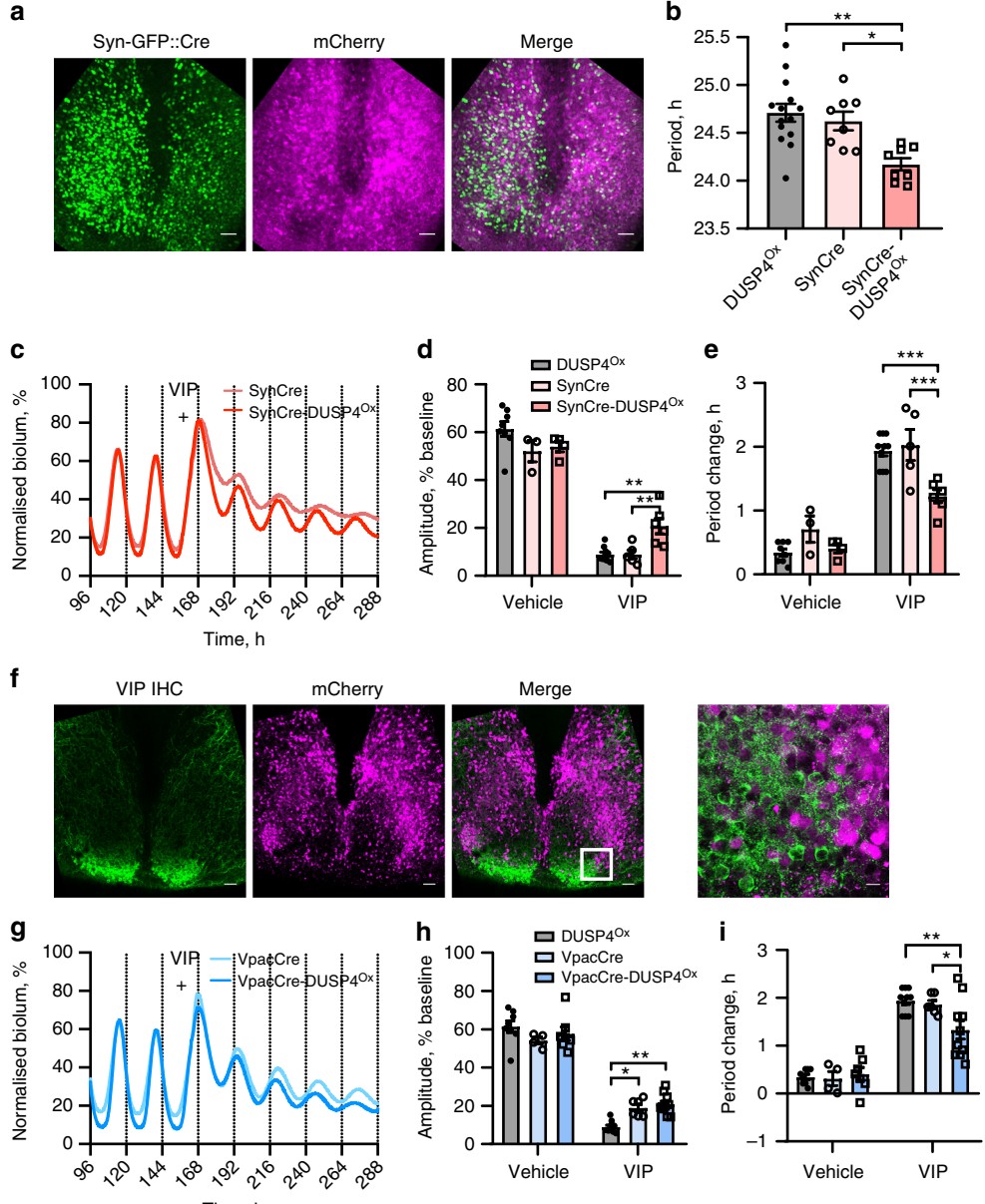

**Fig. 8** Constitutive DUSP4 expression shortens circadian period and reduces the VIP response in SCN slices. **a** Representative confocal micrographs of an SCN slice transduced with *hSyn.GFP::Cre* AAV and *Ef1a.DiO.mCherry-P2A-Dusp4* (SynCre-DUSP4$^{Ox}$). **b** Group data for steady-state (pre-treatment) period of slices transduced as in **a**, along with individual AAV controls (mean ± SEM, one-way ANOVA, $n = 14$ for *DUSP4$^{Ox}$* group, $n = 8$ *SynCre* and *SynCre-DUSP4$^{Ox}$*). **c** Representative PER2::LUC bioluminescence rhythms of *SynCre* and *SynCre-DUSP4$^{Ox}$* slices treated with 1 μM VIP (marked by plus). **d, e** Group data for post-treatment relative amplitude (**d**), and period change (**e**) responses (mean ± SEM) of slices treated with vehicle or VIP at CT10. $n$ as follows: *DUSP4$^{Ox}$*: veh, 8; VIP, 10; *SynCre*: veh, 3; VIP, 5; *SynCre-DUSP4$^{Ox}$*: veh, 4; VIP, 6. **f** Representative confocal micrographs of a *Vpac2-Cre* SCN slice transduced with *Ef1a.DiO.mCherry-P2A-Dusp4* (*VpacCre-DUSP4$^{Ox}$*), immunostained for VIP to demonstrate separate populations of VIP and VPAC2 neurons (close up on right, white boxed region). **g** Representative PER2::LUC bioluminescence rhythms of *VpacCre* and *VpacCre-DUSP4$^{Ox}$* slices treated with 1 μM VIP (marked by plus). **h, i** Group data for post-treatment relative amplitude (**h**), and period change (**i**) responses (mean ± SEM) of slices treated with vehicle or VIP at CT10. *DUSP4$^{Ox}$* data replicated from **d, e**. $n$ as follows: *VpacCre*: veh, 4; VIP, 6; *VpacCre-DUSP4$^{Ox}$*: veh, 7; VIP, 10. All tests in **d, e, h, i** were two-way ANOVAs with Sidak's multiple comparisons. *$P < 0.05$, **$P < 0.01$, ***$P < 0.001$

SCN network level[64]. It is also possible that the desynchronisation between intracellular genetic and cytoplasmic oscillators adds a further level of plasticity that facilitates shifting at the cell-autonomous level.

We identified ERK1/2 and its negative regulator DUSP4 as key components of the VIP transduction cascades, thereby extending previous studies[65–67]. DUSP4 appears to have several roles within

the circadian system, whereby manipulation of DUSP4 levels altered the steady-state SCN oscillation, and responses to VIP and to light. The attenuated responses to delaying light pulses of DUSP4-null mice, in which ERK signalling will be disinhibited, are consistent with the abolition of phase shifting in mice expressing a constitutively active ERK1/2 activator[68]. The ERK1/2 and DUSP4 interaction in the shell is analogous to the CRTC1-SIK1 pathway in

the core[39], and both may serve to limit the response of the SCN to external stimuli to prevent maladaptive changes. An unexpected finding was the altered periodicity when DUSP4 was either over-expressed or deleted. Previous studies have found no such involvement of ERK1/2 in period determination[68–70], although it is important to note that DUSP4 regulates other MAPK pathways concomitantly, including JNK1/2/3 and p38; JNK cascades have been shown to influence the stability of core clock proteins PER2 and BMAL1 and affect period[71,72] (an action supported by our own data). DUSP4 may, therefore, exercise roles in SCN circadian time-keeping beyond VIP-dependent plasticity.

Our results shed light on the mechanisms through which VIP functions both to maintain circuit coherence, and to act as a second-order link between the retinorecipient SCN core and the distal shell. Foremost are the effects of VIP on the cell-autonomous clockwork, while the ability of VIP to re-programme the components of the TTFL may be central to its circadian effects. At both an intracellular level and the network level, the SCN is able to exhibit a high degree of VIP-induced plasticity that is an important feature of resetting mechanisms.

## Methods

**Mouse strains**. All animal work was licensed under the UK Animals (Scientific Procedures) Act 1986, with Local Ethical Review by the MRC. All relevant ethical regulations were complied with. Mouse breeding and genetics were performed by members of the transgenic mouse facility, Ares. All mice older than P14 were culled by a Schedule 1 method: dislocation of the neck followed by exsanguination. All pups P14 and younger were culled by decapitation. DUSP4$^{-/-}$ mice were obtained from the European Mouse Mutant Archive and were subsequently crossed with the PER2::LUC, generated by Dr. Joseph Takahashi[28] (University of Texas Southwestern Medical Center, Dallas). *Dusp4$^{flx/flx}$* were generated by crossing the DUSP4$^{-/-}$-PER2::LUC mouse with a Flp recombinase mouse, resulting in the removal of lacZ and neomycin cassettes, leaving loxP sites located either side of exon 3 of the *Dusp4* gene. Per1-Luc transgenic mice were obtained from Dr. Hitoshi Okamura[45] (Kyoto University). Cry1-Luc transgenic mice were generated in-house and validated by our laboratory[44]. CRY double knockout mice (CryDKO) were bred in-house using *Cry1$^{-/-}$* and *Cry2$^{-/-}$* mice generated by Dr. Gijsbertus van der Horst (Erasmus University, Rotterdam)[73]. The *Vpac2-Cre* line (obtained from JAX Laboratories, Maine) carries a BAC transgene containing Cre recombinase under the control of the *Vipr2* gene promoter.

**Behavioural activity recording**. Mice were individually housed in a light-controllable cabinet for the duration of behavioural monitoring. Their activity patterns were assessed using running wheels (Actimetrics) and passive infrared movement detectors. Mice were typically entrained to a 12:12 LD cycle (which mimicked that of their holding room) for at least 5 days before being exposed to different lighting schedules, including constant dim red light (DD), 8 h phase advances or delays, and Aschoff Type II light pulses (30 min, 400 lux). Food and water were provided ad libitum. Wheel-running data were acquired and stored as wheel revolutions per six-minute bin. Data were analysed using ClockLab (ActiMetrix Inc.) to calculate behavioural period by Chi-squared periodogram, activity duration (alpha), and activity onsets and offsets. Phase angles of entrainment, defined as the difference in time between lights off and activity onset, were calculated based on these onsets in Microsoft Excel. 50% phase shift values (PS50s) were calculated by fitting a sigmoidal curve to onset (phase advance) or offset (phase delay) values using the interpolate function in Graphpad Prism.

**Organotypic slice culture**. SCN organotypic slices were prepared as described[74]. Slices were cultured for at least seven days before any bioluminescence recording. For long-term slice maintenance, slices were kept at 37 °C, 5% CO$_2$ with a medium change every 10 days. For bioluminescent recording, slices were transferred to 35 mm culture dishes containing 1.2 ml HEPES-buffered medium with 100 μM luciferin[74] (Biosynth), and were then air-sealed using glass coverslips secured with silicon grease. Bioluminescence produced by firefly luciferase (in PER2::LUC, Per1-Luc or Cry1-Luc slices) was measured by photomultiplier tubes (Hamamatsu) housed above culture dish stages within a retractable light-tight sleeve in a light-tight incubator kept at 37 °C. Photons were registered every second (counts per second, cps) and were integrated over 6 min bins. Time-lapse imaging of bioluminescence was performed using a charge-coupled device (CCD) camera (Hamamatsu Orca II) and inverted microscope with heated stage contained within custom-built light-tight housing. Images were acquired at 1 frame per hour, and compiled in FIJI (ImageJ) post-acquisition. An Olympus LV200 Luminoview was used for combined bioluminescence and fluorescence imaging. All images were acquired at a rate of 2 frames per hour, with exposure times of 1 ms for brightfield, 200 ms for fluorescence and $1.75 × 10^6$ ms (~30 min) for bioluminescence using a C9100-13 EM-CCD camera (Hamamatsu).

**Viral transduction of slices**. For adeno-associated viruses (AAVs), SCN slices were given a medium change immediately prior to AAV addition. A volume of 1 μl of AAV was placed directly on to the SCN slice and left for one week to allow expression before subsequent recording or further treatment. For lentiviruses (provided by Dr. Marco Brancaccio), SCN slices were treated two days after dissection by dropping 5 μl lentivirus directly onto the SCN slice. Superfluous residuum was washed off after two days.

**Pharmacological treatment of slices**. VIP (Tocris) and Bay 55-9837 (Tocris) stocks were prepared by dissolving in HEPES-buffered medium. Glutamate (1 mM; Tocris) was dissolved in 1eq. NaOH, Tetrodotoxin (TTX, 1 μM; Sigma) and BDNF (200 ng/ml; R&D Systems) were dissolved in water. BDNF was applied at CT16, while TTX was applied 24 h prior to VIP treatment. Sotrastaurin (300 nM; Cayman Chemical), SP600125 (3 μM; Tocris), SCH772984 (100 nM; Cayman Chemical) and 666-15 (1 μM; Tocris) were all dissolved in DMSO, Rp-8-Br-cAMPS (50 μM; Biolog) in water, while PKI 14-22 amide myristoylated (1 μM; Tocris) was dissolved in 30% acetonitrile; these 6 drugs were used 30 min prior to VIP treatment. All pharmacological agents were bath-applied to SCN slices unless otherwise stated, and washed off only if stated explicitly. Phase of treatments was extrapolated from the preceding rhythm. Due to their arrhythmic nature, *Cry1$^{-/-}$Cry2$^{-/-}$* slices were treated based on *Cry1$^{-/-}$Cry2$^{+/-}$* littermate phases (data not shown).

**Bioluminescence analysis**. Bioluminescence recordings from adult and pup slice cultures were analysed to calculate circadian period length, amplitude and relative amplitude error (RAE; a measure of the robustness of a rhythm) using the Fast Fourier Transform–Non-Linear Least Squares (FFT-NLLS) function in the Bio-Dare software[75]. Changes in these values following drug treatment, rather than the absolute values themselves, were often used to account for intrinsic variability between SCN slices. Unless otherwise stated, values were calculated based on rhythms four days before and four days after treatment excepting the first 36 h immediately after treatment application. Phase shifts were calculated as follows: the pre-treatment recording was used to predict when subsequent bioluminescence peaks would occur by extrapolating multiples of the measured period to the last identified peak before the treatment. These were compared with the actual observed peaks of bioluminescence following treatment and any differences were calculated in hours. PER2::LUC induction was calculated by identifying the bioluminescence values for three peaks prior to drug treatment, and extrapolating what the next peak would have been using the 'growth' function in Microsoft Excel. The actual bioluminescence value of the peak following the drug treatment was compared to this predicted value, and a fold-change could then be calculated. If this approach was not possible (for example for CryDKO slices), induction of bioluminescence was calculated instead by subtracting the normalised bioluminescence value immediately before treatment from the highest value within the 6 h recording window after treatment for each slice. Also for CryDKO slices, root mean square (RMS; a measure of how much a signal deviates from zero) was calculated as a replacement for amplitude as there is no reliable PER2::LUC oscillation in CryDKO slices. Raw *Cry1$^{-/-}$Cry2$^{-/-}$* bioluminescence traces were first normalised to the highest value within each dataset (to account for absolute gain differences between PMTs), before performing a 24 h rolling baseline subtraction so that the signal oscillated around zero. Values 48 h before treatment and 48 h after treatment (allowing a 24 h interval which was not included immediately post-treatment) were squared, and the mean for each interval was calculated. The square root of these means was taken and a percentage change between the two intervals was calculated for each slice.

**Time-lapse image analysis**. The Semi-automated route for image analysis (SARFIA)[35] package for Igor Pro software (v. 6.3; Wavemetrics) was used for time-lapse region of interest (ROI) and centre of mass (CoM) analysis. ROIs within the slice were identified following despeckling and background subtraction in FIJI by thresholding the slice images and optimising pixel limits using SARFIA. The positions of these ROIs, their individual bioluminescence intensity profiles and raster plots generated from these profiles were then produced in Igor Pro. Subsequent circadian analysis of ROIs using BioDare was performed as described. Rayleigh plots displaying phase coherence could be produced from these data by converting phase information of each ROI into circular data using Microsoft Excel, and plotted using the R 'circular' package. For CoM analysis[34], time-lapse series were processed (despeckled, background subtracted and normalised) and thresholded in FIJI before using an in-house plugin for Igor Pro from which co-ordinates were generated.

**Microarray**. Organotypic SCN slices were divided into four treatment groups for subsequent RNA extraction and microarray analysis to determine the acute effects of VIP treatment. The CT10 group was untreated and would serve as a baseline comparison; CT12 Vehicle slices were treated with vehicle (HEPES-buffered medium) at CT10 and harvested at CT12 (the peak of PER2::LUC bioluminescence); CT12 VIP slices were treated with 10 μM VIP at CT10 and harvested at CT12; CT16 VIP slices were treated with 10 μM VIP at CT10 and harvested at CT16 (the approximate new peak of PER2::LUC following VIP treatment based on previous experiments). Following total RNA extraction, samples were sent to

Cambridge Genomic Services (Department of Pathology, University of Cambridge) for further processing and microarray analysis. Here the total RNA was checked on an Agilent Bioanalyzer 2100 for integrity, with all samples demonstrating an RNA Integrity Number (RIN) of > 9 apart from two (RIN 7.1 and 8.0). Total RNA was amplified using the Ovation Pico WTA v2 kit (NuGEN Technologies), and the resulting cDNA was fragmented and biotinylated using the BiotinIL kit (NuGEN Technologies). Concentration, purity and integrity of this labelled cDNA were determined using the Nanodrop ND-1000 (Thermo Scientific) and by Bioanalyzer before the cDNA was hybridised to a MouseWG-6 v2 BeadChip overnight. After washing, hybridisation was visualised by staining with streptavidin-Cy3 and scanned using a Bead Array Reader (Illumina).

**Microarray analysis.** Raw data were loaded into R using the Bioconductor lumi package[76] and segregated into pairs (such as CT12 Vehicle vs. CT10) for later comparison. These subgroups were then filtered using the detection $p$-value from Illumina to exclude any non-expressed probes, defined as not being significantly different ($p > 0.01$) from negative controls. Data were then transformed using the Variance Stabilisation Transformation (VST)[77] and then normalized using quantile normalisation to remove technical variation between arrays. A global normalisation to allow visualisation of all groups was also performed. The limma package[78] was used for comparisons between sample groups, and results were corrected for multiple testing using False Discovery Rate (FDR) correction. Correlations and clustering of the datasets from each sample were then performed to identify potential outliers and assess the quality of the data. A heatmap was generated using Morpheus software (Broad Institute, software.broadinstitute.org/morpheus/), and clustering was carried out using the in-built Hierarchical Clustering function in Morpheus. Transcripts found to be significantly regulated by VIP were examined using the DAVID ontology software[79,80] to identify significantly regulated pathways (using gene ontology (GO) terms) through the functional annotation chart function. These significant GO terms were then clustered and visualised using the enrichment map plugin[81] for Cytoscape (v 3.5)[82]. Only terms significant at $P < 0.01$ and $q < 0.05$ were visualised. Significantly regulated genes were analysed for the presence of cAMP/Ca$^{2+}$ regulatory elements (CREs) in promoters by comparison with the dataset from Zhang et al.[43]. Transcripts that were not found in that dataset were excluded from this analysis. Once the circadian or CRE regulation of transcripts had been established, significant enrichment (e.g. for CRE regulation amongst VIP-upregulated transcripts) was tested using the chi-squared test. The observed number of genes regulated in a given way was compared with expected values calculated based on the ratios seen when looking at all transcripts in the CT12 VIP vs. CT12 Vehicle dataset.

**Cloning of DUSP4 CRISPR and overexpression constructs.** The CRISPR-Cas9 system[83] was used to knockout *Dusp4* in SCN slices. Guide RNA (gRNA) sequences were cloned into *pAAV-U6sgRNA(SapI)_hSyn-GFP-KASH-bGH* (Addgene #60958)[52], which contains a gRNA and gRNA scaffold sequence under the control of the human U6 promoter (an RNA polymerase III promoter), as well as a GFP reporter localised to the cytoplasm by a Klarsicht, ANC-1, Syne Homology (KASH) domain under the human synapsin promoter. gRNA sequences were designed using either the gRNA Design Tool by Atum (https://www.atum.bio/products/crispr#4) or the MIT CRISPR design tool (http://crispr.mit.edu/) and the 20 nucleotide targets were as follows: g1, GCTGCAATACCATCGTGCGG; g2, GCGCCCTCTTAGCCCTCGCC; g3, ACTGCGTTTTGCCGGCGACA; g4, CTCCATCGTCACCATGTCGC; g5, GGCGGCGGCGGCCAGCGCGGG; g6, AACAAGAAGGAACCAGGCGA. These oligos, along with complementary 20 nt oligomers, were synthesised (by Sigma) and annealed together. Annealing was carried out by mixing 2 μg of each oligomer (dissolved in water at 100 μM) together with its counterpart, then making the final solution up to 50 μl with annealing buffer (10 mM EDTA, 10 mM Tris (pH 8) and 50 mM NaCl). This was heated at 95 °C for 5 min and then allowed to cool to room temperature. The oligomers were designed such that there would be a 3 nt overhang at each end (ACC and AAC at the 5′ and 3′ ends, respectively) so that, following annealing, the double stranded product could be ligated into *pAAV-U6sgRNA(SapI)_hSyn-GFP-KASH-bGH* digested with SapI. Final constructs were validated in cells and successful plasmids were packaged into AAVs. Co-transduction was carried out with *pAAV.Mecp2.SpCas9* (Addgene #60957).

Conditional DUSP4 overexpression was achieved by cloning *Dusp4* cDNA, alongside an mCherry tag, into *pAAV-EF1a-double floxed-hChR2(H134R)-mCherry-WPRE-HGHpA* (Addgene #20297). mCherry-P2A was amplified from an in-house plasmid using the forward primer 5′-TATGGTACCTTCATAGGGCCGGGATTCTCC-3′ and the reverse primer 5′-TATGTCGACCATGGTGAGCAAGGGCGAG-3′, containing KpnI and SalI restriction sites respectively. The product was digested with KpnI and SalI and ligated into a similarly digested *CMV.Dusp4-myc* vector (Origene MR222119). mCherry-P2A-Dusp4 was amplified from the resulting vector using the forward primer 5′-TATGCTAGCCGTCGACCATGGTGAGCAA-3′ and reverse primer 5′-TATGGCGCGCCTACAGCTGGGGGGAGGTGG-3′, which incorporated NheI and AscI restriction sites, respectively. After digestion with NheI and AscI, the product was ligated into *EF1a-double floxed-hChR2(H134R)-mCherry-WPRE-HGHpA* (Addgene #20297), which had been similarly digested, to produce *Ef1a.DiO.mCherry-P2A-Dusp4*. This was validated in cells and subsequently packaged into AAV particles.

**RNA extraction from SCN slices.** SCN slices were washed in sterile PBS before detaching them from the membrane filter and storing them in RNALater (Qiagen) for 1–30 days at 4 °C to preserve the RNA. RNA extraction was carried out with an RNeasy Micro Kit (Qiagen) using an adapted protocol from the manufacturer's recommendation. RNALater surrounding the slice was removed by pipette and replaced with 350 μl Buffer RLT to disrupt and homogenise the tissue, assisted by 30 s of vortexing. One volume of 70% ethanol was added to the lysate and mixed, then transferred to an RNeasy MinElute spin column and centrifuged for 30 s at $12,000 \times g$. a volume of 350 μl Buffer RW1 was added to the column and spun through for 30 s at $12,000 \times g$, before carrying out a DNase I incubation to remove genomic DNA. A volume of 350 μl Buffer RW1 and $2 \times 500$ μl Buffer RPE washes were then performed, again for 30 s at $12,000 \times g$. A final wash with 80% ethanol was done, centrifuging for 2 min at $12,000 \times g$, with any remaining residue being removed by a subsequent 5 min spin at full speed ($\sim21,000 \times g$). RNA was eluted in 14 μl RNase-free water, and its purity and concentration were assessed via Nanodrop. cDNA was subsequently synthesised from RNA using an iScript cDNA synthesis kit (Bio-Rad) according to manufacturers instructions, with the result diluted 1:2 in nuclease-free water before subsequent processing.

**qPCR.** A volume of 15 μl of cDNA solution from each SCN slice was pooled to generate the highest concentration standard. Serial dilutions were then performed to produce the remaining standards. Samples were diluted 1:10 to ensure they fell within this standard curve, and to provide sufficient sample volume for all genes investigated. qPCR was carried out using a Prime Pro 48 machine (Techne) and KAPA SYBR Fast qPCR reagents (KAPA Biosystems). Each sample well consisted of 6.5 μl KAPA SYBR master mix, 0.5 μl of each primer at 10 μM (see Table 1 below) and 2.5 μl nuclease-free water prepared as a master mix, plus 2.5 μl cDNA. Plates were sealed, vortexed and centrifuged briefly, and were then run in the Prime Pro 48 with the following conditions: 95 °C for 2 min, then 40 cycles of 5 s at 95 °C and 30 s at an optimised annealing temperature. At the end of the cycling, a melt curve programme was run which involved heating to 95 °C for 15 s, then dropping to 55 °C for 15 s and gradually increasing in temperature to 95 °C. Analysis was carried out using Prime Pro Study software, excluding wells with unexpected melt curves (suggesting a non-specific PCR product). Samples were compared to the standard curve to determine relative quantity of a given transcript, which was in turn normalised to RNS18 quantity to control for variable total RNA levels. The primers in Table 2.5 were made by Sigma and validated in-house. In addition to these, pre-validated PrimePCR SYBR Green Assay primers were purchased from Bio-Rad for mouse *Mat2a* and *Cxcl10* transcripts. For *Mat2a*, a 147 bp amplicon was amplified from the 115–291 bp region of the Mat2a transcript. For *Cxcl10*, a 61 bp amplicon was amplified from the 608-698 bp region of the *Cxcl10* transcript. Both primers used 60 °C as their annealing temperature.

**X-gal section staining.** Following careful dissection, brains were immediately post-fixed in 10 ml 4% paraformaldehyde (PFA; Alfa Aesar) made up in phosphate buffer (108 mM Na$_2$HPO$_4$.2H$_2$O, 25.3 mM NaH$_2$PO$_4$.2H$_2$O) and shaken for 2 h at 4 °C before being cryopreserved in 20% sucrose (Fisher Chemical, UK) in PBS at 4 °C overnight. The brains were then mounted on a freezing microtome using OCT embedding medium (Thermo Scientific) and 40 μm coronal sections were taken, rostral to caudal. Sections were placed in wells of WHO dimple trays containing 0.01 M PBS. Solution A (0.43 mg ml$^{-1}$ X-gal, 2% N,N-dimethylformamide, 1 mM MgCl$_2$; in PBS) was mixed with Solution B (21 mg ml$^{-1}$ potassium hexacyanoferrate (II) trihydrate, 16.5 mg ml$^{-1}$ potassium hexacyanoferrate (III); in H$_2$O) at a ratio of 9:1 to make X-gal staining solution. Following washes in PBS, the sections were incubated in X-gal staining solution at 37 °C in a humidified chamber for 12–48 h. The sections were washed in PBS and then mounted onto slides before being allowed to air dry. The slides were briefly rinsed in water to remove residual salts and then dehydrated in 95% ethanol and 100% ethanol for 2 min each. Slides were placed in Clear-Rite 3 (Thermo Scientific) for 2 min and finally coverslipped using Pertex Mounting Medium (HistoLab). Images were acquired on an Olympus BX41 microscope with a Nikon DS2mv camera attachment.

**Transient transfection of cells.** Neuroblastoma2A cells (N2As; PHE culture collection 88112303) were cultured in DMEM with Glutamax (Invitrogen) supplemented with 10% FCS in a 37 °C humidified incubator at 5% CO$_2$. For transient transfection, cells were seeded in 12-well plates at a density of $1 \times 10^5$ one day prior to transfection. A volume of 1 μg total plasmid DNA was used, incubated with 3 μl FuGene 6 Transfection Reagent (Promega) in 250 μl OptiMEM (GIBCO) per well for 20 min. This transfection solution was then gently added to each well. Cells were typically harvested 72 h after transfection for western blots. For downstream fixation and imaging, cells were seeded in 24-well plates on poly-L-lysine-coated 13 mm coverslips at a density of $5 \times 10^4$ per well one day prior to transfection and were transfected with FuGene 6, OptiMEM and 0.5 μg total DNA using the same ratios and protocol as stated above. *Ef1a.DiO.mCherry-P2A-Dusp4* was co-transfected with *pAAV.CAG.iCRE-2A-H2BGFP.WPRE* (kind gift of Ernesto Ciabatti, MRC, LMB). Cells were fixed after 72 h.

**Western blotting.** N2A cells were washed in warm PBS and harvested in 100 μl RIPA buffer supplemented with one cOmplete, EDTA-free Protease Inhibitor

**Table 1 Primers used for qPCR of genes to validate microarray data**

| Gene name | Primer direction | Sequence | Amplicon length (bp) | Annealing temperature (°C) | Primer pair source |
|---|---|---|---|---|---|
| Adamts1 | Foward | AGTGGTGTGTCAGTGGCAAG | 166 | 65 | 85 |
|  | Reverse | TTCTTTGGGACTGGGTTGTC |  |  |  |
| Cartpt | Foward | AAGAAGTACGGCCAAGTCCC | 86 | 59 | Primer BLAST |
|  | Reverse | CAGTCACACAGCTTCCCGAT |  |  |  |
| Dbp | Foward | AATGACCTTTGAACCTGATCCCGCT | 175 | 58 | 86 |
|  | Reverse | GCTCCAGTACTTCTCATCCTTCTGT |  |  |  |
| Dusp4 | Foward | CTGTACCTCCCAGCACCAAT | 233 | 65 | 87 |
|  | Reverse | GACGGGGATGCACTTGTACT |  |  |  |
| Irs2 | Foward | GCGGCCTCATCTTCTTCACT | 130 | 58 | 88 |
|  | Reverse | AACTGAAGTCCAGGTTCATATAGTCAGA |  |  |  |
| Per1 | Foward | CACCACTGCCGATCTAAAGC | 186 | 59 | In-house |
|  | Reverse | TCGAGGGGAGAATACTGGGA |  |  |  |
| Per2 | Foward | CCTACAGCATGGAGCAGGTTGA | 94 | 58 | In-house |
|  | Reverse | TTCCCAGAAACCAGGGACACA |  |  |  |
| Ptgs2 | Foward | TGAGTACCGCAAACGCTTCT | 148 | 59 | Primer BLAST |
|  | Reverse | ACGAGGTTTTTCCACCAGCA |  |  |  |
| Rn18s | Foward | CGCCGCTAGAGGTGAAATTC | 62 | 58 | In-house |
|  | Reverse | TTGGCAAATGCTTTCGCTC |  |  |  |
| Vgf | Foward | AGGAGGAGGACGGGGAAG | 135 | 65 | 89 |
|  | Reverse | TCTGCGGATCCATCTCCTC |  |  |  |

Cocktail (Roche) tablet per 10 ml. Samples were subsequently sonicated to ensure cell lysis and to shear DNA before centrifugation at $21,000 \times g$ at 4 °C for 10 min to obtain the soluble protein fraction. Equal amounts of protein was denatured for 5 min at 90 °C in the presence of 4x NuPAGE LDS Sample Buffer (Invitrogen) and 25 mM DTT. Samples were loaded onto 4–12% Bis-Tris NuPage gels (Invitrogen) alongside a Novex Sharp Pre-Stained Protein Standard (Invitrogen) and run using the MOPS buffer system at 200 V for 45 min. Proteins were transferred using a wet tank system at 30 V for 1 h at 4 °C onto polyvinylidene difluoride (PVDF) membranes (Immobilon-P Membrane, 0.45 μm, Millipore) activated in methanol. Membranes were subsequently blocked for 1 h at room temperature in 5% (w/v) non-fat dried milk (Marvel) in TBS-T and incubated with anti-myc tag (1:2000, Abcam ab9106) overnight at 4 °C. Membranes were then washed three times in TBS-T for 15 min each and incubated in anti-rabbit IgG coeroxinjugated to horseradish pdase (CST 7074). After three more 20 min washes in TBS-T, the ECL Prime (GE Healthcare) chemiluminescence system was used for detection, imaged using a GelDoc system (Bio-Rad) and analysed in Image Lab software (Bio-Rad) Rabbit anti-β-actin (1:5000, CST 4967) was used as a loading control.

**Immunohistochemistry**. Freshly dissected brains were fixed in 4% PFA for 3 h at room temperature, then cryopreserved in 20% sucrose/PBS overnight at 4 °C. The brains were then mounted on a freezing microtome using OCT embedding medium and 40 μm coronal sections containing SCN were taken, rostral to caudal. Sections were permeabilised in 100% methanol for 10 min at −20 °C, washed in PBS and then blocked in blocking buffer (2% normal goat serum, 1% BSA, 0.3% triton X-100 in PBS) for 1 h shaking at room temperature. Primary antisera incubation was carried out overnight at 4 °C using rabbit anti-pERK (CST, 197G2) at 1:1000 in antibody buffer (1% BSA, 0.3% triton X-100 in PBS). Cells were then washed in a 1:3 dilution of antibody buffer in PBS before secondary antibody incubation with goat anti-rabbit Alexa fluor 488 (Invitrogen) at 1:500 in the same buffer. Sections were washed and mounted onto Superfrost Plus slides (Thermo Fisher), rinsed in water and coverslipped using Vectashield Hardset mounting medium with DAPI (Vector labs). Confocal imaging was performed using a Zeiss 780 microscope and a 20x objective, with fluorescence measurements performed on FIJI.

Immunohistochemistry of SCN slices was performed as above with some modifications. Slices were fixed for 1 h at 4 °C 30 min after drug treatment. Permeabilisation, blocking and primary antisera incubation were performed as above, but washes were carried out in undiluted antibody buffer over several days after secondary antibody incubation. Confocal imaging was performed using a Zeiss 780 microscope and ×20 or ×63 objectives.

Immunocytochemistry of cells was performed using the above procedure with some modifications. Cells were fixed for 20 min at 4 °C, there was no methanol permeabilisation step, primary antisera incubation was carried out using rabbit anti-myc tag (Abcam, ab9106) at 1:1000, and the secondary antibody used was goat anti-rabbit Alexa fluor 647 (Invitrogen) at 1:500. Cells were washed, rinsed in water and the coverslips transferred to slides. Images were acquired using a Nikon HCA inverted fluorescence microscope.

**AAV production**. HEK293T cells were seeded in 15 cm plates at a density of $5 \times 10^6$ one day prior to transfection. Cells were co-transfected with 7 μg AAV 2/1 (Rep/Cap proteins, for serotype 1), 20 μg pHGTI-Adeno1 (adenoviral helper) and 7 μg of the vector plasmid itself per plate, using polyethylenimine (PEI) in a 1:4 ratio DNA:PEI, mixed with 5 ml of DMEM. About 6 h after transfection, the medium was changed, with the harvest beginning 72 h post-transfection. The supernatant from each plate was harvested (and pooled if applicable) before centrifugation in a tabletop centrifuge at $2000\times g$ for 30 min. The resultant supernatant was then filtered through a 0.45 μm Steriflip (Millipore) and transferred to polyallomer 30 ml ultracentrifuge tubes (Beckman Coulter). A volume of 2 ml of sterile 20% sucrose/PBS was gently pipetted to the bottom of these tubes before ultracentrifugation at $40,000 \times g$ at 4 °C for 2 h (Optima XPN-90 Ultracentrifuge, Beckman Coulter). The supernatant was subsequently removed and discarded, leaving the AAV pellet. The polyallomer tubes were allowed to air dry before the pellet was resuspended in 25 μl sterile, ice-cold PBS. This was left on ice for 1 h before combining resuspensions (if necessary), aliquotting and storing at −80 °C. Other AAVs were purchased commercially: *AAV8.hSyn.GFP::Cre* (UNC Vector Core); *AAV1.CAG.DiO-tdTomato* and *AAV1.Syn.NES.jRCaMP1a. WPRE.SV40* (Penn Vector Core)[84].

**General analysis**. Statistical tests and graphical representation of data (typically mean ± SEM) were performed using Prism 6 or 7 software (Graphpad). Two-tailed, unpaired Student's *T*-tests were performed in experiments containing two groups separated by one independent variable. More than two groups separated by one independent variable were tested using an Ordinary one-way ANOVA, with Tukey correction for multiple comparisons where every group is compared to every other group, or Dunnett correction where groups are compared to a control group. Two-way ANOVAs with Tukey correction for multiple comparisons between all groups were performed on experiments involving multiple independent factors (e.g. time and drug treatment). If no significant differences are found between individual groups, then any significant differences between factors are reported. Pearson's correlation coefficient was calculated to determine if two variables were correlated and, if applicable, linear regression was used to see if one variable could be predicted from another.

**Biological materials**. All unique materials (plasmids, gRNA sequences, qPCR primers, viral vectors) are available from the authors, excluding: two pre-validated qPCR primers for Mat2a and Cxcl10, which are commercially available from BioRad, and two commercially available AAVs: AAV8.hSyn.GFP::Cre (UNC Vector Core); AAV1.Syn.NES.jRCaMP1a.WPRE.SV40 (Penn Vector Core).

**Reporting summary**. Further information on experimental design is available in the Nature Research Reporting Summary linked to this article.

## Data availability

Microarray data have been deposited in the Gene Expression Omnibus under accession code GSE113797. The datasets generated during the current study are

available from the corresponding author on reasonable request. A reporting summary for this Article is available as a Supplementary Information file.

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

## Acknowledgements

All animals were cared for in accordance with the UK Animals (Scientific Procedures) Act of 1986 with local ethical approval. We thank the Biomedical Facilities staff at the MRC-LMB Ares facility for mouse breeding and handling. We also thank Cambridge Genomic Services, who performed the microarray experiments and initial data processing. This work was funded by the Medical Research Council (MC_U105170643 to MHH).

## Author contributions

R.H. and M.H.H. designed research; P.C. provided reagents; R.H. and J.E.C. performed research; R.H. analysed data; and R.H. and M.H.H. wrote the paper.

## Additional information

**Competing interests:** The authors declare no competing interests.

