## [Peer Review File · Nature Communications]

Reviewers' comments:

Reviewer #1 (Remarks to the Author):

In this manuscript, Hamnett and Hastings use a range of sophisticated animal models and approaches to determine intracellular signaling mechanisms that underpin actions of vasoactive intestinal polypeptide (VIP) in the suprachiasmatic nuclei (SCN), site of the brain's main circadian pacemaker. Neuropeptides are key to the functioning of the SCN, and VIP acting via its cognate VPAC2 receptor is arguably the most important. Over the past 20 years or so, pharmacological, physiological, and genetic investigations have firmly implicated VIP-VPAC2 signaling in coupling autonomous cellular oscillators in the SCN as well as in the resetting of the SCN by environmental signals. However, little is known regarding how intercellular signaling by VIP is transduced by SCN cells. Here the authors show that VIP given at CT10 acts in intact as well as in TTX-impaired SCN networks to phase delay rhythms in PER2::LUC and that these effects are mimicked by a VPAC2 agonist. These actions accompanied by acute induction in PER2::LUC, the magnitude of which is correlated with the size of the delay. Further, VIP evoked a long-lasting increase in period of PER2::LUC oscillations. Such changes were not seen when VIP was given at CT22, indicating phase-dependency in these actions. Subsequently the authors used microarrays to identify genes that were up- or down-regulated by VIP given at CT10. They identified clock genes as well as *Dusp1* and *Dusp4* as being robustly induced by VIP at this time. GO term analysis also indicated that MAPK pathways were potentially recruited by VIP at this time. They then show that CREs were present in many VIP-regulated genes including those in MAPK pathways. Interestingly LUC reporters of Ca²⁺ or CRE activity were differentially affected VIP with their circadian relationships disrupted post-VIP treatment. Similarly, VIP differentially altered different LUC reports of key circadian clock genes, alteration in *Per1*-Luc being quite different to the that PER2::LUC or *Cry1*-Luc. Then, through pharmacological means, it is shown that inhibitors of ERK1/2 reduce resetting responses in PER2::LUC to VIP. Finally, the authors show through genetic and other means that *Dusp4*, a negative regulator of ERK1/2, is a component of the downstream signaling mechanisms of VIP in the SCN. In general the statistical analysis seems appropriate and there are many aspects of this work that will find a broad audience in neuroscience, and in particular, circadian neurobiology and peptide pharmacology. There are several novel findings in this manuscript and for the most part, it all hangs together, however there are some points for the authors to consider.

1) The examination of VIP effects in this study are primarily related to its phase-delaying actions when given at the late subjective day (CT10). The authors draw parallels with photic resetting, since VIP neurons are known to be innervated by retinal ganglion cells. However, it is unclear how VIP effects parallel those of light since light pulses typically evoked large delays CT12-16 and not CT10. Why did they not also examine the resetting responses of glutamatergic stimuli at this time? Since they note that the SCN reprogrammed *in vitro*, perhaps the phase-response curve to glutamate is also altered?

2) The observations that VIP re-programs the SCN is intriguing. Does this mean then that current concepts of the relations among the different intracellular signaling apparatus and the TTFL are incorrect? Or are factors present *in vivo* that resist such reprogramming? Or that these relationships are stable across several different solutions?

3) Related to 2), did the authors examine whether the reprogramming noted above was reversible with subsequent media changes or is the long-term robustness of the rhythm compromised by the re-programming i.e. does the rhythm dampen out more quickly?

4) Minor point. The phase advancing effects of light in Figure 6h indicated considerably variability for +/+ and +/- mice--how was this data analyzed? Non-parametrics would seem most appropriate, or, dropping the animals with the large advances since they are robust outliers. It seem more accurate to state that at this transition phase, light really does not evoked reliable

advances in any strain or that in the absence of *Dusp4*, variability is reduced.

5) Minor point. The present findings firmly implicate MAPK pathways in VIP's actions. Several studies indicate an overt rhythm in activated ERK1/2 in the SCN and in the absence of VIP signaling, this *de novo* expression of activated ERK1/2 is diminished in the SCN (Hughes et al 2004). This suggests that intrinsic VIP signaling also recruits MAPK pathways in the SCN.

6) Minor point: In the Methods(line 660), it states that 'myself' did something. I think they mean RH.

Reviewer #2 (Remarks to the Author):

The authors have prepared a tour-de-force analysis of VIP's effects on the SCN. While the quantity of experiments included is impressive, it is the quality and diversity of approaches that is truly exceptional. The authors have been so thorough that there really are few if any open questions. This study will be of incredible use to scientists interested in the mammalian circadian system, not only for finely detailing how VIP works in the SCN, but also for providing a template on how to investigate such issue using a vast array of techniques, from behavior, to *in vitro* reporters, to viral manipulation of genes, to the used of CRISPER-CAS9 to modify genes in wildtype tissue. The discovery and analysis of *DUSP4*'s participation is exciting as it provides an interesting new target for future investigation. The manuscript has been so expertly and thoughtfully prepared that I have relatively few constructive comments, all of which are minor.

1) In figure 4AC and text on line 212, it is stated that the period in the CRE-luc slices is unchanged by VIP treatment. Visualizing the period in the VIP is impossible for the reader, as the y-axis is so large to accommodate the massive induction by VIP. Is it possible to include an inset with a more modest axis that would allow the reader to see the oscillation?

2) in the figures that include an amplitude change plot (e.g., Fig 2f,j; Fig5a, fig 8f), the y-axis is not intuitive. These are labeled as "Amplitude change, %". The treatments that have the biggest effects have the tiniest bars, implying the smallest changes according to the axis label. Something like "Amplitude, % baseline" would be more accurate.

3) in figure 4, the "+" on traces 1,b,d and j are nicely labeled. the same should be done for the "+" in figures 7b and 8b, and a "+" should be included in figure 1A.

4) Line 258-260 - This is a little confusing as written. PKA and PKC were not examined. Rather, their roles were explored using specific inhibitors.

5) Line 261 - it is stated that "These inhibitors had no measurable effect ... when applied individually" however examining fig S5f-i it is clear that these inhibitors on their own had a measurable, but not significant, effect. The effects size is actually quite large, and the sample sizes quite small (n=4-5), leading me to worry that the authors are making a Type 2 error claiming that there is no effect. The lack of statistical details here makes this null finding rather hard to interpret. I'd recommend rephrasing this section somewhat to more accurately reflect the situation (a non-significant increase on their own, but a significant increase when together).

5) line 481 - Bioluminescence is misspelled.

Reviewer #3 (Remarks to the Author):

In this paper, the authors report pharmacological and genetic studies of the mechanisms by which neuropeptide vasoactive intestinal peptide (VIP) affects the mammalian central circadian clock, the suprachiasmatic nucleus (SCN). In SCN slices, bath application of VIP induced acute induction of the clock protein PER2, phase shift of the PER2 circadian rhythm, and a profound disruption of its orderly spatiotemporal pattern. The acute induction and phase shift effects were preserved when action potentials were blocked by tetrodotoxin (TTX), suggesting that they result from cell-autonomous mechanisms. Assuming this is true, the rest of the paper explores effects of VIP in SCN slices by using gene expression microarrays, optical reporters of CRE/Ca/Cry1/Per1 rhythms, and pharmacological blockers of JNK/ERK, and effects of VIP in SCN slices or locomotor activity rhythms using mice deficient in or overexpressing the JNK/ERK disinhibitor DUSP4. The main conclusion is that effects of VIP on SCN are mediated at least partly through ERK1/2 and regulated by DUSP4. This is a beautifully written and illustrated paper presenting a large number of experiments employing advanced technology, including single-cell imaging of SCN PER2/Per1/Cry1/Ca/CRE rhythms, microarrays to assess gene expression effects of VIP, and CRISPR-Cas9 manipulations of DUSP4 in mice.

Major Concerns:

A) Bath application of VIP to SCN slices bypasses important features of natural VIP signaling in SCN, i.e. restricted spatial localization, synaptic release, temporal/circadian patterning, and co-release of other transmitters. This concern is mitigated somewhat by the TTX results in Fig. 2 and the elegant co-culture studies of Maywood et al., but still substantially limits the physiological relevance of these studies.

B) The main conclusion about the role of ERK1/2 signaling in response of SCN to VIP rests solely on pharmacological experiments in Fig. 5, as DUSP4 is not specific to ERK vs. JNK (Kidger, Semin Cell Dev Bio 50:125, 2016).

C) In the study using DUSP4 knockout mice, use of the VPAC2-Cre line to restrict DUSP4 knockout to only those neurons receiving VIP input would provide stronger evidence that DUSP4 is important for VIP signaling in SCN.

Minor Comments:

1) In Fig. 1a, add "+" symbol. In Fig. 1h, increase size of inset. In Fig. 4, place panels d/e in the same row, and f/g/h/i in the next row. In Figs. 5d & 8f, change vertical axis labels from "Amplitude change" to just "Amplitude". In Fig. 5, show PER2::LUC traces with inhibitors only. In Fig. S8c images, use red for mCherry, blue for DAPI, and green for CRE:GFP.

2) In SCN slice experiments, were drugs always allowed to remain for the entire duration of the experiment? Define "angle of entrainment".

3) Cite McCarthy, Eur Neuropsychopharmacol 26:1310, 2016.

4) Delete "briefly" (line 133). Change "in isolation" to "alone" (line 246). The word "however" is not a conjunction and cannot join two independent clauses (lines 246, 338, 430).

VIP + Dusp4 paper rebuttal

We are grateful for the constructive comments and explicit corrections provided by the Reviewers. We have addressed all of the conceptual and technical points raised, as demonstrated below, and all typographic errors have been corrected. **Yellow shows where we have made significant amendments to the text.**

Reviewer 1

1) The examination of VIP effects in this study are primarily related to its phase-delaying actions when given at the late subjective day (CT10). The authors draw parallels with photic resetting, since VIP neurons are known to be innervated by retinal ganglion cells. However, it is unclear how VIP effects parallel those of light since light pulses typically evoked large delays CT12-16 and not CT10. Why did they not also examine the resetting responses of glutamatergic stimuli at this time? Since they note that the SCN reprogrammed in vitro, perhaps the phase-response curve to glutamate is also altered?

The Reviewer is correct in highlighting this potential ambiguity between light, glutamate and VIP. We set out to study the mechanisms through which VIP signalled in the SCN in relation to its phase resetting effects, but it is important to note, as we now seek to emphasise in the introduction, that the role of VIP in the SCN is not merely a mimic of the acute response to light. In the text we now emphasise the differences between primary afferent, glutamatergic light-dependent cues, and the second-order, core-to-shell signalling mediated by VIP onto VPAC2 positive cells. This pathway will undoubtedly progress photic regulation in vivo, but it is also critical for ongoing SCN circuit coherence, ensemble phase and ensemble period determination under free-running conditions in vivo and in slices in vitro.

We have also taken up the Reviewer's suggestion of comparing VIP directly with glutamate, conducting a new set of experiments that reveal the distinct molecular effects and temporal dependence of the two resetting cues. The results of these new experiments are in Supplementary Fig. S2.

In text: **To compare the phase-shifting action of VIP with that of glutamate, the primary mediator of RHT photic input to the SCN core, we applied VIP or glutamate via droplet directly on to SCN slices at either CT10 or CT14. Whereas VIP caused significant phase-shifts at both times, glutamate caused a phase-shift only at CT14 (Supplementary Fig. S2g-i), consistent with previous reports based on the circadian**

cycle of electrical firing and PER2::LUC^{31,32}. Equally, VIP acutely induced PER2 at both time points but glutamate did so only at CT14 (Supplementary Fig. S2j). Furthermore, glutamate did not result in a subsequently reduced amplitude at either phase, in contrast to VIP (Supplementary Fig. S2k). Thus, the effects of VIP on the SCN clock network are distinct from those of glutamate (and by extension light), in terms of their phase-dependence and molecular consequences^{23,32}.

2) *The observations that VIP re-programs the SCN is intriguing. Does this mean then that current concepts of the relations among the different intracellular signaling apparatus and the TTFL are incorrect? Or are factors present in vivo that resist such reprogramming? Or that these relationships are stable across several different solutions?*

This is an equally intriguing set of questions. First, we are not yet able, for technical reasons, to speculate or translate our findings to an in vivo context because the necessary technology for precise, high-resolution longitudinal measures of gene expression, cellular calcium, CRE-transcription etc. are still being developed. We cannot say, therefore, whether factors present in vivo facilitate or impede the re-programming we see in the slice, although the echoes between VIP in vitro and LL in vivo are tempting and we have briefly mentioned this in the revised text. On the point as to whether “*current concepts of the relations among the different intracellular signaling apparatus and the TTFL are incorrect*” we take this to ask about the relation between TTFL and the cytosolic rhythms in, for example, Ca²⁺, cAMP etc. Our opinion is that the simple high-level view that “cytosolics drive the TTFL and the TTFL drives the cytosolics” is correct, but of course the Devil is in details. Our empirical observation that sustained VIP signalling can re-align the sub-components of these two systems does show that a stable steady state can be sustained by at least two solutions. A formal mathematical analysis is currently not available but would be an ideal future way to determine the “landscape” of solutions possible. It is likely the case that VIP-mediated photoperiodic encoding for seasonal responses will be one realistic setting in which multiple context-specific solutions may be exploited by the SCN. We have addressed this directly in the text.

In results: **The long-term persistence of this VIP-induced state is reminiscent of the long period and loss of rhythm definition and amplitude seen in mice exposed to constant light (LL)²⁹, a condition that would continuously excite VIP neurons.**

In discussion: VIP activity does not, therefore, mimic the effects of light pulses and glutamate, although the effects of pharmacological treatment with VIP are reminiscent of constant light, in which VIP cells would be continuously stimulated. The VIP-mediated period-lengthening and reduced definition of cell-autonomous and network rhythms of the TTFL revealed here at the level of the SCN may thus be the cause of the corresponding changes seen in circadian behaviour.

3) Related to 2), did the authors examine whether the reprogramming noted above was reversible with subsequent media changes or is the long-term robustness of the rhythm compromised by the re-programming ie does the rhythm dampen out more quickly?

We have addressed this query directly by adding new experimental data. Remarkably, the re-programming is not reversible by repeated media-changes. The rhythm remains highly stable, notwithstanding its low amplitude, for the duration of our recording, which can exceed 20 days. We have included these new data in Supplementary Fig. S1c-e.

4) Minor point. The phase advancing effects of light in Figure 6h indicated considerably variability for +/+ and +/- mice--how was this data analyzed? Non-parametrics would seem most appropriate, or, dropping the animals with the large advances since they are robust outliers. It seem more accurate to state that at this transition phase, light really does not evoked reliable advances in any strain or that in the absence of *Dusp4*, variability is reduced.

This is fair comment. Originally the data were analysed as described in the figure legend, with a one-way ANOVAs with Tukey's multiple comparisons test, but on the Reviewer's advice we have re-analysed it with the non-parametric Kruskal-Wallis test with Dunn's multiple comparisons correction. Also following the Reviewer's advice, we have rephrased the text as follows: In contrast, no significant difference between genotypes was observed for an advancing light pulse (30 min, 400 lux) delivered at ZT22 (Fig. 6h), although advances for all genotypes at this phase were not robust

5) Minor point. The present findings firmly implicate MAPK pathways in VIP's actions. Several studies indicate an overt rhythm in activated ERK1/2 in the SCN and in the absence of VIP signaling, this de novo expression of activated ERK1/2 is diminished in

the SCN (Hughes et al 2004). This suggests that intrinsic VIP signaling also recruits MAPK pathways in the SCN.

We thank the Reviewer for this additional support that VIP is likely to signal via the ERK1/2 pathway, and have included the following in the Discussion: We have demonstrated that the cellular signalling cascade involved in transducing VIP/VPAC2 activation to gene expression is the MAPK pathway, specifically ERK1/2, consistent with the observation that pERK rhythmicity is diminished in the absence of VIP signalling⁶⁶.

6) Minor point: In the Methods(line 660), it states that 'myself' did something. I think they mean RH.

Thank you, this has been corrected to in-house.

Reviewer 2

1) In figure 4AC and text on line 212, it is stated that the period in the CRE-luc slices is unchanged by VIP treatment. Visualizing the period in the VIP is impossible for the reader, as the y-axis is so large to accommodate the massive induction by VIP. Is it possible to include an inset with a more modest axis that would allow the reader to see the oscillation?

We agree that our plot in Fig. 4a is unsuitable. We have now included a detrended oscillation, plotted on the re-scaled right-hand y-axis, that allows the oscillation to be seen. Hopefully this improves the Figure.

2) in the figures that include an amplitude change plot (e.g., Fig 2f,j; Fig5a, fig 8f), the y-axis is not intuitive. These are labeled as "Amplitude change, %". The treatments that have the biggest effects have the tiniest bars, implying the smallest changes according to the axis label. Something like "Amplitude, % baseline" would be more accurate.

This is a fair point - we had not considered that particular interpretation of the axis label. All axis labels have now been changed to avoid the ambiguity.

3) in figure 4, the "+" on traces 1,b,d and j are nicely labeled. the same should be done for the "+" in figures 7b and 8b, and a "+" should be included in figure 1A.

Thank you, we have added appropriate labels (e.g. "VIP") to graphs of representative traces.

4) Line 258-260 - This is a little confusing as written. PKA and PKC were not examined. Rather, their roles were explored using specific inhibitors.

Agreed. We have added in the following words in yellow to clarify this: To confirm further the specific contribution of the ERK1/2 pathway to the VIP response, **the involvement of** two other kinases frequently implicated in circadian phase-resetting, protein kinase A (PKA)²⁵ and protein kinase C (PKC)⁴⁸, was also tested **pharmacologically**

5) Line 261 - it is stated that "These inhibitors had no measurable effect ... when applied individually" however examining fig S5f-i it is clear that these inhibitors on

their own had a measurable, but not significant, effect. The effects size is actually quite large, and the sample sizes quite small (n=4-5), leading me to worry that the authors are making a Type 2 error claiming that there is no effect. The lack of statistical details here makes this null finding rather hard to interpret. I'd recommend rephrasing this section somewhat to more accurately reflect the situation (a non-significant increase on their own, but a significant increase when together).

Thank you, we have reworded this to replace “measurable” with “significant”: These inhibitors had no **significant** effect on the response to VIP when applied individually (Supplementary Fig. S7e-h).

It may be that we explained the figure in a misleading way. To explore this further below, we have boxed the 3 relevant bars, which do not appear (to the naked eye, as well as statistically) different to the black control bar on the left.

Reviewer 3

A) Bath application of VIP to SCN slices bypasses important features of natural VIP signaling in SCN, i.e. restricted spatial localization, synaptic release, temporal/circadian patterning, and co-release of other transmitters. This concern is mitigated somewhat by the TTX results in Fig. 2 and the elegant co-culture studies of Maywood et al., but still substantially limits the physiological relevance of these studies.

We thank the Reviewer for mentioning our earlier “paracrine signalling” studies, which definitely have a bearing on how we think about VIP and the SCN circuit. We fully agree that bath application is a simplified system. In particular, the half-life of VIP in static culture is considerably longer than *in vivo* (An et al., 2011). We have now included this caveat in our discussion. We have also added new experimental data, presented in Supplementary Fig. S1f-j, which used time-limited treatment with VIP to test the time-frame over which its effects are established. We applied VIP as previously but this was followed by a wash-off, the timing of which was designed to mirror the microarray time points (i.e. 2 h and 6 h after VIP application). These data confirm that the duration of exposure time to VIP is an important determinant of the overall response, although some components of the VIP response, such as PER2 induction, amplitude reduction and baseline increase, are seen when VIP is present for only 2 h (and in spite of any effects that media change may have).

In the text, we have described these experiments as follows: **The emergence of these effects was, however, progressive, with reduced amplitude established by only 2 h of VIP treatment, whereas sustained period lengthening required more than 6 h of VIP (Supplementary Fig. S1e-j).**

We have also added the following to the Discussion: **It is recognised that the bath application of VIP to slices used here is a simplified system that may not reflect all aspects of endogenous VIP signalling, such as duration of VIP exposure. Nevertheless, previous studies have revealed very clearly an important role for diffuse, paracrine signalling of VIP⁸, and when VIP was applied here for shorter time periods, its effects (PER2 induction, phase shifting, amplitude reduction) were still observed and developed progressively.**

B) The main conclusion about the role of ERK1/2 signaling in response of SCN to VIP rests solely on pharmacological experiments in Fig. 5, as DUSP4 is not specific to ERK vs. JNK (Kidger, Semin Cell Dev Bio 50:125, 2016).

Thank you for this suggestion, we have now included a sentence in the Results section when first introducing the DUSP family that they are capable of

dephosphorylating multiple MAP kinase proteins: we focussed on *Dusp4*, which can dephosphorylate MAP kinase proteins such as ERK1/2 and JNK1/2/3⁵¹, as a potential regulator of the effects of VIP.

To provide further (non-pharmacological) experimental evidence that VIP acts through phosphorylated ERK1/2 as suggested, we undertook some immunostaining studies to determine if VIP signalling increased pERK levels. These new data are presented in Supplementary Fig. S6, in which it can be seen that VIP strongly upregulates pERK compared to vehicle application.

This experiment was also performed in *VpacCre*-TdT_o slices, in which VPAC2 cells are fluorescent through a floxed TdT_o gene transduced via AAV. Here, strong pERK signal can be seen in VPAC2 cells, providing evidence that not only is VIP acting through the ERK1/2 pathway, but that it is doing so specifically in VIP-receptive VPAC2-positive cells (additional evidence for this can be seen in point C addressing DUSP4 manipulation in VPAC2 cells).

In text: Further evidence for the involvement of ERK1/2 came from immunostaining of SCN slices, which showed that VIP treatment at CT10 increased phosphorylated ERK1/2 (pERK) signal (Supplementary Fig. S6b,c). Moreover, by exploiting intersectional expression of TdT_o fluorescent reporter driven in a VPAC2-Cre mouse line, VIP-induced pERK signal could be clearly localised to VPAC2-positive cells (Supplementary Fig. S6d).

C) In the study using DUSP4 knockout mice, use of the VPAC2-Cre line to restrict DUSP4 knockout to only those neurons receiving VIP input would provide stronger evidence that DUSP4 is important for VIP signaling in SCN.

We fully agree and thank the Reviewer for the suggestion. Unfortunately, the *VpacCre* allele is expressed in the germ line and so deletes floxed alleles during gametogenesis (we discovered this the hard way in a different experimental programme). Consequently, we cannot create a mouse with selective homozygous deletion of *Dusp4* solely in VPAC2 cells. Hence, we have instead adopted an AAV-mediated approach to overexpress DUSP4 specifically the VPAC2 neurons in SCN slice culture, which we hope is an acceptable alternative to show that DUSP4 is important specifically in the VPAC2 neurons. The data from these new experiments are presented in Fig. 8 and Supplementary Fig. S10. They demonstrate that the effects of pan-neuronal overexpression of DUSP4 on the period-lengthening response to VIP are replicated when DUSP4 is over-expressed in just the VPAC2 neurons.

In text: To focus on a role for DUSP4 in VPAC2-positive cells alone, DUSP4 was overexpressed specifically in VPAC2-Cre SCN slices (Fig. 8f,g). Unlike when DUSP4 was constitutively expressed in all neurons, expression in VPAC2 neurons did not affect steady state period (Supplementary Fig. S10g,h), and with respect to the VIP response, it had no significant effect on acute PER2 induction or phase shifting (Supplementary Fig. S10i,j). Amplitude reduction was reduced relative to DUSP4^{Ox} slices, albeit not compared to VpacCre controls (Fig. 8h). Deletion of Dusp4 specifically from VPAC2-positive cells did, however, significantly attenuate the VIP-induced period-lengthening (Fig. 8i).

Minor Comments:

1) In Fig. 1a, add "+" symbol. In Fig. 1h, increase size of inset. In Fig. 4, place panels d/e in the same row, and f/g/h/i in the next row. In Figs. 5d & 8f, change vertical axis labels from "Amplitude change" to just "Amplitude". In Fig. 5, show PER2::LUC traces with inhibitors only. In Fig. S8c images, use red for mCherry, blue for DAPI, and green for CRE:GFP.

We thank the Reviewer for these suggested improvements to our Figures. All have been incorporated, apart from the "Amplitude change" axis label, which has been changed to "Amplitude, % baseline" as suggested by Reviewer 2.

2) In SCN slice experiments, were drugs always allowed to remain for the entire duration of the experiment? Define "angle of entrainment".

That is correct, unless explicitly stated that they were washed off. For clarity, we have added the following text to the Methods section: All pharmacological agents were bath-applied to SCN slices unless otherwise stated, and washed off only if stated explicitly

We have now defined phase angle of entrainment in the Methods section: Phase angles of entrainment, defined as the difference in time between lights off and activity onset, were calculated based on these onsets in Microsoft Excel

3) Cite McCarthy, *Eur Neuropsychopharmacol* 26:1310, 2016.

We thank the Reviewer for this suggestion and we have included this citation as an additional example of ERK1/2 and DUSP proteins in circadian regulation: We then identified DUSP proteins, a family previously implicated in circadian regulation^{39,67}, as VIP-sensitive

4) Delete “briefly” (line 133). Change “in isolation” to “alone” (line 246). The word “however” is not a conjunction and cannot join two independent clauses (lines 246, 338, 430).

Thank you, all suggestions have been incorporated.

REVIEWERS' COMMENTS:

Reviewer #1 (Remarks to the Author):

In this revised version of their manuscript, Hamnett and colleagues present compelling evidence that VIP's shifting actions on the SCN circadian clock depend in part on ERK1/2 and DUSP4 signaling. The authors use a range of cutting edge approaches to both replicate and consolidate earlier findings in the field that used more conventional methods and to implicate DUSP4 as new intracellular factor in VIP's remodeling of the SCN. The authors have addressed concerns raised and have set out some really intriguing findings which illustrate unanticipated plasticity and novel stable solutions for relationships among intracellular signaling and the TTFL.

Reviewer #2 (Remarks to the Author):

The authors have expertly addressed all of my concerns and comments. This is a wonder series of experiments and will make a substantial contribution to the field.

Reviewer #3 (Remarks to the Author):

My concerns have been addressed in this outstanding revision.